# Angle resolved photoemission spectroscopy reveals spin charge separation in metallic MoSe$_2$ grain boundary

Yujing Ma[1], Horacio Coy Diaz[1], José Avila[2,3], Chaoyu Chen[2,3], Vijaysankar Kalappattil[1], Raja Das[1], Manh-Huong Phan[1], Tilen Čadež[4,5], José M.P. Carmelo[4,5,6], Maria C. Asensio[2,3] & Matthias Batzill[1]

Material line defects are one-dimensional structures but the search and proof of electron behaviour consistent with the reduced dimension of such defects has been so far unsuccessful. Here we show using angle resolved photoemission spectroscopy that twin-grain boundaries in the layered semiconductor MoSe$_2$ exhibit parabolic metallic bands. The one-dimensional nature is evident from a charge density wave transition, whose periodicity is given by $k_F/\pi$, consistent with scanning tunnelling microscopy and angle resolved photoemission measurements. Most importantly, we provide evidence for spin- and charge-separation, the hallmark of one-dimensional quantum liquids. Our studies show that the spectral line splits into distinctive spinon and holon excitations whose dispersions exactly follow the energy-momentum dependence calculated by a Hubbard model with suitable finite-range interactions. Our results also imply that quantum wires and junctions can be isolated in line defects of other transition metal dichalcogenides, which may enable quantum transport measurements and devices.

[1] Department of Physics, University of South Florida, Tampa, Florida 33620, USA. [2] Synchrotron SOLEIL, L'Orme des Merisiers, Saint Aubin-BP 48, Gif sur Yvette Cedex 91192, France. [3] Université Paris-Saclay, L'Orme des Merisiers, Saint Aubin-BP 48, Gif sur Yvette Cedex 91192, France. [4] Beijing Computational Science Research Center, Beijing 100193, China. [5] Center of Physics of University of Minho and University of Porto, Oporto P-4169-007, Portugal. [6] Department of Physics, University of Minho, Campus Gualtar, Braga P-4710-057, Portugal. Correspondence and requests for materials should be addressed to J.M.P.C. (email: carmelo@fisica.uminho.pt) or to M.C.A. (email: asensio@synchrotron-soleil.fr) or to M.B. (email: mbatzill@usf.edu).

1D electron systems (1DES) are sought for their potential applications in novel quantum devices, as well as for enabling fundamental scientific discoveries in materials with reduced dimensions. Certainly, 1D electron dynamics plays a central role in nanoscale materials physics, from nanostructured semiconductors to (fractional) quantum Hall edge states[1,2]. Furthermore, it is an essential component in Majorana fermions[3,4] and is discussed in relation to the high-$T_c$ superconductivity mechanism[5]. However, truly 1D quantum systems that permit testing of theoretical models by probing the full momentum-energy ($k$, $\omega$)-space are sparse and consequently angle-resolved photoelectron spectroscopy (ARPES) measurements have only been possible on quasi-1D materials consisting of 2D- or 3D-crystals that exhibit strong 1D anisotropy[6–10].

Electrons confined in one-dimension (1D) behave fundamentally different from the Fermi-liquid in higher dimensions[11–13]. While there exist various quasi-1D materials that have strong 1D anisotropies and thus exhibit 1D properties, strictly 1D metals, that is, materials with only periodicity in 1D that may be isolated as a single wire, have not yet been described as 1D quantum liquids. Grain boundaries in 2D van der Waals materials are essentially 1D and recent DFT simulations on twin grain boundaries in $MoS_2$ (ref. 14) and $MoSe_2$ (ref. 15) have indicated that those defects should exhibit a single band intersecting the Fermi level. Therefore, such individual line defects are exceptional candidates for truly 1D metals.

In the case of quasi-1D Mott-Hubbard insulators (MHI)[16–19], there is strong evidence for the occurrence of the so called spin-charge separation[17,18]. Recently, strong evidence of another type of separation in these quasi-1D compounds was found, specifically a spin-orbiton separation with the orbiton carrying an orbital excitation[16].

The theoretical treatment of MHI is easier compared with that of the physics of 1D metals. The ground state of a MHI has no holons and no spinons and the dominant one-electron excited states are populated by one holon and one spinon, as defined by the Tomonaga Luttinger liquid (TLL) formalism[12]. For 1DES metals the scenario is however more complex, as the holons are present in both the ground and the excited states. Zero spin-density ground states have no spinons. Consequently, the experimental verification of key features of 1DES, especially the spin-charge separation, remains still uncertain[6,20–22].

The theoretical description of 1DES low-energy excitations in terms of spinons and holons, based on the TLL formalism, has been a corner stone of 1D electron low-energy dynamics[12]. The rather effective approximation of the relation of energy versus momentum in 1D fermions by a strictly linear dispersion relation, makes the problem accessible and solvable, by calculating analytically the valuable many-body low-energy dynamics of the system. This drastic assumption has provided an effective tool to describe low-energy properties of 1D quantum liquids in terms of quantized linear collective sound modes, named spinons (zero-charge spin excitations) and holons (spinless charge excitations), respectively. However, this dramatic simplification is only valid in the range of low-energy excitations, very close to the Fermi level.

More recently, sophisticated theoretical tools have been developed that are capable to extend this description to high-energy excitations away from the Fermi-level[13,23–28]. Particularly, the pseudofermion dynamical theory (PDT)[24–27] allows to compute one-particle spectral functions in terms of spinon and holon features, in the full energy versus momentum space (($k$, $\omega$)-plane). The exponents controlling the low- and high-energy spectral-weight distribution are functions of momenta, differing significantly from the predictions of the TLL if applied to the high-energy regime[23–27]. To the best of our knowledge, while other theoretical approaches, beyond the TLL limit, have also been recently developed[13,28], no direct photoemission measurements of spin-charge separation in a pure metallic 1DES has been reported so far. Even more important, a theoretical 1D approach with electron finite-range interactions entirely consistent with the photoemission data in the full energy versus momentum space has never been reported before[11,12,29].

Here we present a description of the non-Fermi liquid behaviour of a metallic 1DES with suitable finite-range interactions over the entire ($k$, $\omega$)-plane that matches the experimentally determined weights over spin- and charge-excitation branches. This has been accomplished by carrying out the first ARPES study of a 1DES hosted in an intrinsic line defect of a material and by developing a new theory taking electron finite-range interactions within an extended 1D Hubbard model into account. The mirror twin boundaries in a monolayer transition metal dichalcogenide[30,31] are true 1D line defects. They are robust to high temperatures and atmospheric conditions, thus making them a promising material system, which is amendable beyond ultra high vacuum investigations and useful for potential device fabrication. Previously, the structural properties of these line defects have been studied by (scanning) transmission electron microscopy[15,30–32] and by scanning tunnelling microscopy (STM) and tunnelling spectroscopy[33–35].

## Results

**Line defect characterization.** Figure 1 shows STM results of the mono- to bilayer $MoSe_2$ grown on a $MoS_2$ single crystal substrate. Three equivalent directions for the MTBs are observed in the hexagonal $MoSe_2$ crystal. The high density of these aligned line defects in $MoSe_2$ (ref. 30) provides a measurable ARPES signal for this 1DES and thus enables the $\omega(k)$ characterization of this line defect.

**Peierls transition in $MoSe_2$ grain boundary.** For metallic 1D structures, an instability to charge density wave (CDW) is expected (see additional discussion in Supplementary Note 1), which has been previously reported for $MoSe_2$ grain boundaries by low temperature STM studies[35]. The CDW in MTBs gives rise to a tripling of the periodicity, as can be seen in the low temperature-STM images shown in Fig. 2a,b. The CDW in 1D metals is a consequence of electron-phonon coupling. The real-space periodicity of the CDW is directly related to a nesting of the Fermi wavevector, as schematically shown in Fig. 2c. ARPES measurements of the Fermi-surface can thus directly provide justification for the periodicity measured in STM, which is shown below. In addition, the CDW transition is a metal-insulator transition and thus changes in the sample resistance occur at the CDW transition temperature. Figure 2d shows a four-point measurement with macroscopic contacts on a continuous mono- to bi-layer film (as shown in Fig. 1c). Clear jumps in the resistance are observed for three different samples at $\sim 235 K$ and $\sim 205 K$, which are attributed to an incommensurate and commensurate CDW transitions, respectively. The drop in resistance at lower $T$ is assigned to a depinning of the CDW from defects and so-called CDW sliding. CDW sliding is a consequence of the applied potential rather than a specific temperature.

To study a stable, gapless, 1DES, we determine the spectral weight together with the energy dispersion in momentum space, by performing ARPES measurements at room temperature, which is well above the CDW transition temperature. This is done on samples consisting predominantly of monolayer $MoSe_2$ islands, as shown in the Supplementary Fig. 1. Figure 3; Supplementary Fig. 2 illustrate the Fermi surface of 1D metals, consisting of two

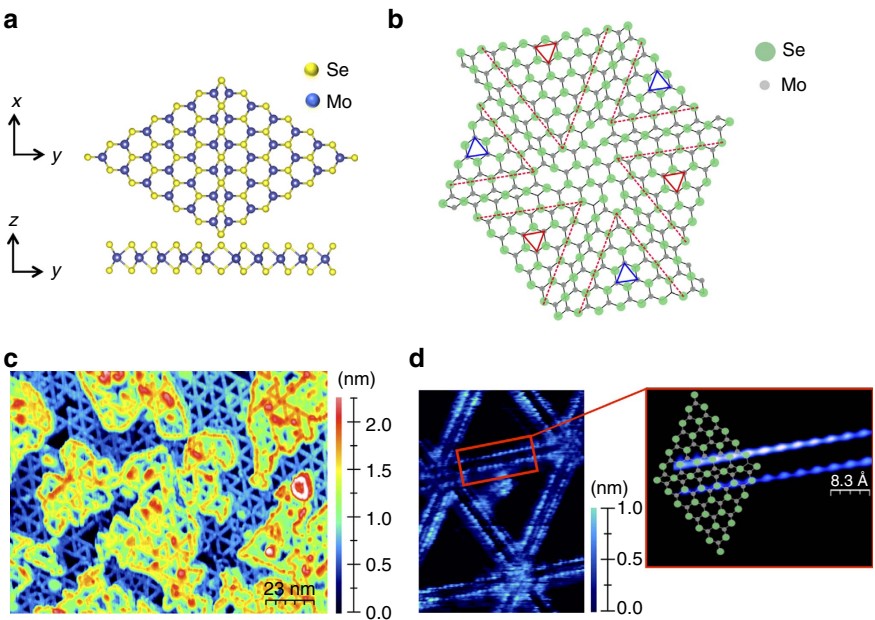

**Figure 1 | Defect structure of mirror twin grain boundaries (MTBs) in monolayer MoSe₂.** (**a**) Ball-and-stick model of a MTB, indicating that the grain boundary is Se deficient. (**b**) Arrangements of the three equivalent MTB directions gives rise to a cross-hatched grain boundary network. (**c**) Large-scale (150 × 110 nm²) STM image of 1–2 monolayers of MoSe₂ grown by MBE on MoS₂. The MTBs appear as bright lines forming a dense network of aligned line defects. In higher resolution images shown in (**d**) the defect lines appear as two parallel lines. Imaging at room temperature allows resolving atomic corrugation along these lines that are attributed to atom positions in the Se-rows adjacent to the defect line, as the overlay of the model illustrates.

parallel lines, separated by $2k_F$, in the absence of interchain hopping. Because of the three equivalent real space directions of the MTBs in our sample, super-positioning of three rotated 1DES results in star-shaped constant energy surface in reciprocal space, as shown in Fig. 3; Supplementary Note 1. In the three cases, a perfect nesting is noticeable, namely one complete Fermi sheet can be translated onto the other by a single wave vector $\pm 2k_F$.

Even more important, by using high energy and momentum resolution ARPES, the Fermi-wave vector could be precisely determined, giving a value of $k_F = 0.30 \pm 0.02\,\text{Å}^{-1}$, which is about 1/3 of the BZ-boundary at $\pi/a_{MoSe_2}$. Hence a band filling of $n = 2/3$ has been experimentally obtained. The Fermi-wavevector also gives a direct prediction of the CDW periodicity of $\pi/k_F = 10.5 \pm 0.7\,\text{Å}$, which is in good agreement with $3 \times a_{MoSe_2}$ measured in STM (Fig. 2).

**Spin charge separation.** While the perfect nesting conditions in 1D metals predicts a CDW transition, its occurrence is no proof for 1D electron dynamics. For obtaining evidence of 1D electron dynamics, a detailed analysis of the spectral function and its consistency with theoretically predicted dispersions need to be demonstrated. The photoemission spectral function of the 1D state is shown in Fig. 3e,f. Without any sophisticated analysis and considering only the raw ARPES data, it is evident that the experimental results are in complete disagreement with the single dispersing band predicted by ground state DFT simulations[15,35]. Effectively, our data cannot be fit with a single dispersion branch (see also Supplementary Fig. 4 and Supplementary Note 2 for an analysis of the raw data in terms of energy distribution curves (EDC), momentum distribution curves (MDC) and lifetime.)

Using data analysis that applies a curvature procedure to raw data[36], as commonly used in ARPES, the experimental band dispersions in the full energy versus momentum space show two clear bands that exhibit quite different dispersions. We provisionally associate, which our theoretical results confirms below, the upper and lower dispersion with the spinon and the

holon branch, respectively. Manifestly, the spin mode follows the low-energy part of the 1D parabola, whereas the charge mode propagates faster than the spin mode. The extracted experimental velocity values are $v_h = 4.96 \times 10^5\,\text{ms}^{-1}$ and $v_s = 4.37 \times 10^5\,\text{ms}^{-1}$, revealing a ratio $v_h/v_s$ of the order of $\approx 0.88$. Notice that these states lie entirely within the band gap of the MoSe₂ monolayer, whose VBM is located at 1.0 eV below the Fermi-level, see Supplementary Fig. 3.

DFT simulations cannot predict the electron removal spectrum of the 1D electron dynamics. Thus the single dispersing band obtained in previous DFT simulations for this system is not expected to be consistent with the experiment. However, the single-band DFT results indicate that the electron dynamics behaviour can be suitably described by a single band Hubbard model and associated PDT. The PDT is a method that has been originally used to derive the spectral function of the 1D Hubbard model in the vicinity of high-energy branch-line singularities[24–27]. It converges with TLL for low energies[37]. As reported below, here we use a renormalized PDT (RPDT) because the conventional 1D Hubbard does not include finite-range interactions.

**Low energy properties and TLL electron interaction strength.** Critical for calculating the spectral functions with RPDT is the knowledge of the electron interaction strength, which needs to be determined experimentally. Since very close to the Fermi level, in the low-energy excitations limit, the RPDT converges to the TLL theory, we have evaluated the photoemission weight in the vicinity of the Fermi-level in accordance to TLL theory. A decisive low-energy property of 1D metals is, according to that theory[12,38], the suppression of the DOS at the Fermi-level, whose power law exponent is dependent on the electron interaction range and strength. Figure 4 shows the angle integrated photoemission intensity, which is proportional to the occupied DOS, as a function of energy for the 1DES. It is compared with the photoemission from a gold sample under the same conditions.

 3

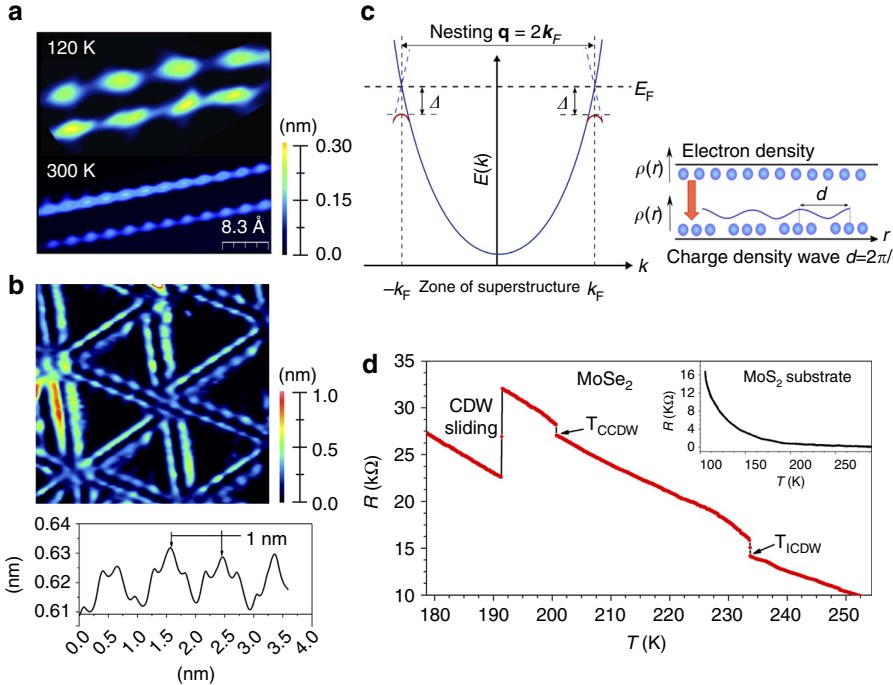

**Figure 2 | Charge density wave (CDW) transition in MTBs.** (**a**) STM images of a single MTB at low temperatures (120 K) exhibit three times the periodicity than the atomic corrugation imaged at room temperature. In (**b**) a larger scale low-T STM image and the corresponding cross-section along the indicated MTB is shown that measured the periodicity of the CDW as ∼1.0 nm. The schematic in (**c**) illustrates the relationship between CDW period and nesting vector $q = 2k_F$. Also the opening of a band gap at $k_F$ is illustrated. Temperature dependent resistance measurements, shown in (**d**), indicate two CDW transitions. The transitions at 235 and 205 K correspond to incommensurate and commensurate CDW transitions, respectively. Depending on the applied bias voltage we also observe a drop in resistance below the CDW transition temperatures, which is attributed to CDW-sliding. The inset shows the control measurement on a bare $MoS_2$ substrate and shows no transitions.

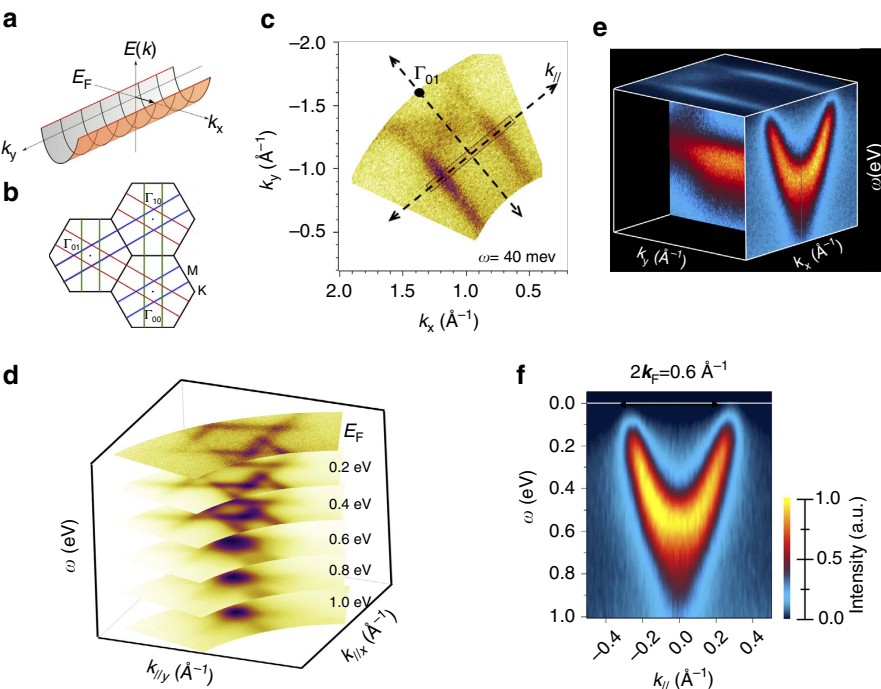

**Figure 3 | ARPES measurement of *k*-space resolved electronic structure of MTBs.** In (**a**) the band structure of a 1D metal is schematically illustrated. The parabolic band disperses in the $k_x$ direction, which is the momentum vector along the 1D defect. The lack of periodicity in the $k_y$ direction causes the replication of the parabola forming a parabolic through and thus the Fermi-surface consists of two parallel lines. In the case of the three equivalent directions of MTBs that are rotated by 120° with respect to each other, three Fermi-surfaces overlap to form the Fermi-surface illustrated in (**b**). The experimental measurement of the Fermi-surface close to the center of the second BZ using left and right circular polarized light is shown in (**d**). By using linear polarized light photoemission from a specific MTB-orientation can be emphasized as shown in (**c**). The Band dispersion $E(k)$ is shown in (**e**) and (**f**) for the momentum slice indicated in (**c**). This momentum slice was chosen because it lies outside of bands for the other two MTB orientations.

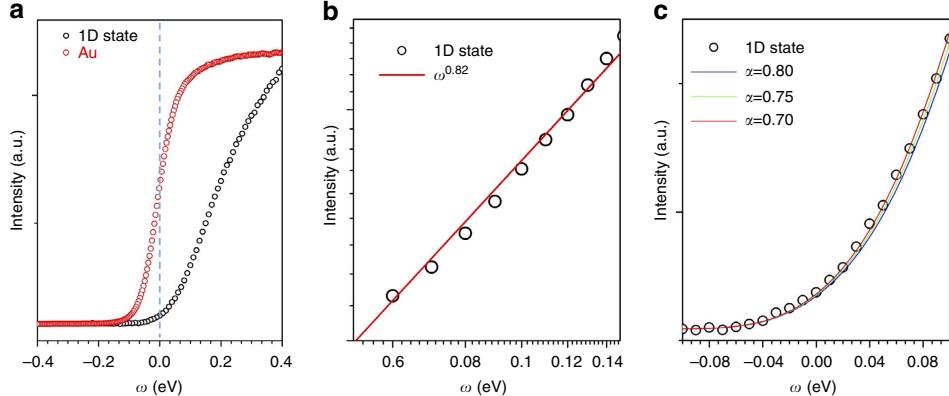

**Figure 4 | Evaluation of the suppression of the density of states at the Fermi level according to TLL theory.** The suppression of the density of states of MTBs close to the Fermi-level compared with the density of states for a regular FL metal (Au) is shown in (**a**), measured at room temperature (to avoid CDW transition). The density of states is obtained by plotting the angle integrated photoemission intensity as a function of binding energy $\omega$. The log plot in (**b**) indicates that the density of states increases[39] with $\omega^{0.8}$, as is shown in (**c**). The data are well fit with $\alpha = 0.75$, but the variation of the fit with the exponent is small and thus the uncertainty in $\alpha$ is estimated to be as large as $\pm 0.05$.

The suppression of the DOS for the 1D defects compared with Au is apparent in Fig. 4a.

According to the TLL scheme, the suppression of DOS follows a power law dependence whose exponent is determined by the electron interaction strength and range in the 1D system. An exponent of $\sim 0.8$ is extracted from a log-plot shown in Fig. 4b. A refined fitting for the exponent $\alpha$ that takes the temperature into account[39] reveals that the data are best reproduced for $\alpha$ between 0.75 and 0.80 (Fig. 4c). The charge TLL parameter $K_c$, which provides information on the range of the electron interaction[29], is related to $\alpha$ by $\alpha = (1 - K_c)^2 / 4K_c$. Hence $K_c$ has values between 0.20 and 0.21.

**Comparison of experiment to the theoretical model.** Within the 1D Hubbard model with on-site repulsion $U$ and hopping integral $t$, the charge TLL parameter $K_c$ and related exponent $\alpha$ values should belong to the ranges $K_c \in [1/2, 1]$ and $\alpha \in [0, 1/8]$, respectively. However, our experimental values are in the ranges $K_c \in [0.20, 0.21]$ and $\alpha \in [0.75, 0.80]$, which is an unmistakable signature of electron finite-range interactions and therefore our system cannot be studied in the context of the conventional 1D Hubbard model[29]. Consequently, we have developed a new theoretical scheme that successfully includes such interactions. As justified below in the Methods section, the corresponding RPDT specifically relies on the spectral function near the branch lines of the non-integrable 1D Hubbard model with finite-range interactions being obtained from that of the integrable 1D Hubbard model PDT[24–27] on suitably renormalising its spectra and phase shifts.

The renormalization using the PDT approach has two steps. The first refers to the $U$ value, which loses its onsite-only character and is obtained upon matching the experimental band spectra with those obtained within the 1D Hubbard model for $n = 2/3$, leading to $U = 0.8t$. Indeed, the ratio $W_h / W_s$ of the observed c band (holon) and s band (spinon) energy bandwidths $W_h = \varepsilon_c(2k_F) - \varepsilon_c(0)$ and $W_s = \varepsilon_s(k_F) - \varepsilon_s(0)$, respectively, is achieved for that model at $U/t = 0.8$. (The energy dispersions $\varepsilon_c(q)$ for $q \in [-\pi, \pi]$ and $\varepsilon_s(q')$ for $q' \in [-k_F, k_F]$ and the related $\gamma = c, c', s$ exponents $\tilde{\zeta}_\gamma(k)$ considered in the following are defined in more detail in the Methods section.) This renormalization fixes the effective $U$ value yet does not affect $t$. The corresponding c and c' (holon) and s (spinon) branch lines spectra $\omega_c(k) = \varepsilon_c(|k| + k_F)$ for $k \in [-k_F, k_F]$, $\omega_{c'}(k) = \varepsilon_c(|k| - k_F)$ for

$\in (-3k_F, 3k_F)$ and $\omega_s(k) = \varepsilon_s(k)$ for $k \in [-k_F, k_F]$ are plotted in Fig. 5d–f; Supplementary Fig. 5. An important difference relative to the $n = 1$ Mott-Hubbard insulating phase is that for the present $n = 2/3$ metallic phase the energy bandwidth $W_c = \varepsilon_c(\pi) - \varepsilon_c(2k_F)$ does not vanish. That the renormalization does not affect $t$ stems from a symmetry that implies that the full c band energy bandwidth is independent of both $U$ and $n$ and reads $W_h + W_c = 4t$. Hence $W_h = 4t$ for the Mott-Hubbard insulator whereas $W_h < 4t$ for the metal. Combining both the value of the ratio $W_h / W_c$ for the 1D Hubbard model at $U/t = 0.8$ and $n = 2/3$ and the exact relation $W_h + W_c = 4t$ with analysis of Fig. 5d–f, one uniquely finds $t \approx 0.58$ eV. The parameter $\alpha$ is here denoted by $\alpha_0$ for the 1D Hubbard model. It reads $\alpha_0 = (2 - \xi_c^2)/(8\xi_c^2) \in [0, 1/8]$ with $\alpha_0 = 0$ for $U/t \to 0$ and $\alpha_0 = 1/8$ for $U/t \to \infty$ where $\xi_c = \sqrt{2K_c}$ is a superposition of pseudofermion phase shifts. (see Methods.)

The second step of the renormalization corresponds to changing the $\xi_c$ and phase shift values so that the parameter $\alpha = (2 - \tilde{\xi}_c^2)^2/(8\tilde{\xi}_c^2)$ has values in the range $\alpha \in [\alpha_0, \alpha_{max}]$ where $\alpha_0 \approx 1.4 \times 10^{-3}$ for $U/t = 0.8$ and $n = 2/3$. As justified in the Methods section, $\alpha_{max} = 49/32 \approx 1.53$. The effect of increasing $\alpha$ at fixed finite $U/t$ and $n$ from $\alpha_0$ to $1/8$ is qualitatively different from that of further increasing it to $\alpha_{max}$. As discussed in that section, the changes in the $(k, \omega)$ plane weight distribution resulting from increasing $\alpha$ within the latter interval $\alpha \in [1/8, \alpha_{max}]$ are mainly controlled by the finite–range interactions.

For $U/t = 0.8$, $n = 2/3$ and $T = 0$ the one-electron spectral function of both the conventional 1D Hubbard model ($\alpha = \alpha_0$) and corresponding model with finite range interactions ($\alpha \in [\alpha_0, \alpha_{max}]$) consists of a $(k, \omega)$-plane continuum within which well-defined singular branch lines emerge. Most of the spectral weight is located at and near such singular lines. Near them, the spectral function has a power-law behaviour characterised by negative $k$ dependent exponents. At $T \approx 300$ K such singular lines survive as features displaying cusps. Our general renormalization procedure leads to a one-electron spectral function expression that for small deviations $(\omega_\gamma(k) - \omega) > 0$ from the finite-energy spectra $\omega_\gamma(k)$ of the $\gamma = c, c', s$ branch lines plotted in Fig. 5d–f reads, $B(k, \omega) \propto (\omega_\gamma(k) - \omega)^{\tilde{\zeta}_\gamma(k)}$ for $\alpha \in (\alpha_0, \alpha_{max})$. The singular branch lines correspond to the $\gamma = c, c', s$ lines $k$ ranges for which their exponents $\tilde{\zeta}_\gamma(k)$ are negative. As confirmed and justified in the Methods section, for $U/t = 0.8$, $n = 2/3$ and $t = 0.58$ eV there is quantitative agreement with the $(k, \omega)$-plane ranges of the experimentally observed spectral function cusps for

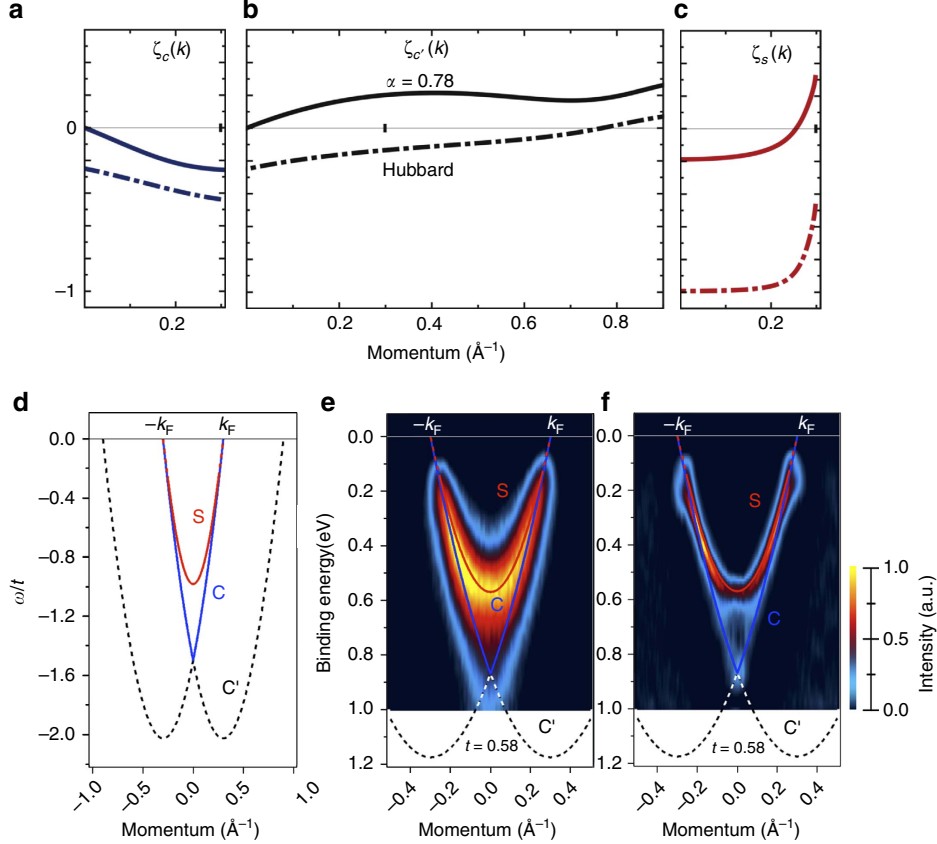

**Figure 5 | Exponents momentum dependence and theoretical and experimental spectral lines.** (**a,b,c**) The exponents that control the spectral function near the c, c′ and s branch lines, respectively, for $U/t = 0.8$, $t = 0.58$ eV and electronic density $n = 2/3$ plotted as a function of $k$ for the 1D Hubbard model with finite-range interactions corresponding to $\alpha = 0.78$ (full lines) and the conventional 1D Hubbard model for which $\alpha_0 \approx 1.4 \times 10^{-3}$ (dashed-dotted lines), respectively. For the former model at $\alpha = 0.78$ the c′ branch line exponent remains positive for all its $k$ range whereas the ranges for which the c and s branch lines exponents are negative coincide with the momentum intervals showing ARPES peaks in (**e**) and (**f**); (**d**) The theoretical c, c′ and s branch line spectra plotted as a function of the momentum $k$ for the 1D Hubbard model with finite-range interactions corresponding to $\alpha = 0.78$ whose full and dashed lines refer to momentum ranges with negative and positive exponents, respectively; (**e**) Energy versus momentum ($k_{//}$) along the $\overline{\Gamma_{01}}\ \overline{K}$ direction in the Brillouin zone, plus the same theoretical lines as in (**d**). The broad spectral line and the spectral continuum between the s and c branch lines apparent in (**e**) are consistent with the behaviour of 1D metals and our theoretical model, see Methods section and supplementary Note 2 for details. The results of applying a curvature procedure to the raw data[36] on panel (**e**) are shown in panel (**f**), together with the theoretically computed c, c′ and s branch lines.

$\alpha \in [0.75, 0.78]$. This is fully consistent with the $\alpha$ experimental uncertainty range $\alpha \in [0.75, 0.80]$. The three $\gamma = c$, $c'$, $s$ exponents momentum dependence for both the 1D Hubbard model with finite-range interactions corresponding to $\alpha = 0.78$ (full lines) and the conventional 1D Hubbard model for which $\alpha = \alpha_0 \approx 1.4 \times 10^{-3}$ (dashed-dotted lines) is plotted in (Fig. 5a–c).

## Discussion
The agreement of the theoretical calculations with finite range interactions over the entire $(k, \omega)$-plane provides strong evidence for the assignment of the two spectral branches observed in the experiments to spin charge separation in a 1D metal. Despite this agreement, alternative explanations for the photoemission spectrum should be noted. Strongly asymmetric line shapes in photoemission spectra have been reported and thus an assignment of the cusps to yet unknown line-shape effects in 1D materials cannot be entirely excluded. However, the accurate prediction of the continuum between the cusp lines and the fit of the c and s branch-line dispersions by the 1D Hubbard model with finite range interactions makes alternative effects unlikely to reproduce exactly such spectral features.

Concerning the DOS at the Fermi level, our measurements clearly show a suppression of the DOS that can be fit with a power law behaviour. DOS suppression has, however, also been observed due to final-state pseudogap effects in nanostructures[40,41]. While it is difficult to exclude such effects categorically, the expected 1D nature of the line defects and thus the breakdown of Fermi-liquid theory requires application of TLL, as has been applied to other (quasi) 1D systems in the past[6,38,42], to interpret photoemission intensity at the Fermi level. Certainly, obtaining the same exponent $\alpha$ for the power law behaviour of TLL from the experimental fit of the DOS and the spectral features of the 1D Hubbard model with finite range interactions support the assignment of the DOS suppression at the Fermi-level to TLL effects.

We have presented a detailed experimental analysis of the electronic structure of a material line defect by angle resolved photoemission. High density of twin grain boundaries in epitaxial monolayer $MoSe_2$ could be analysed by angle resolved photoemission spectroscopy. This enabled us to accurately determine the Fermi surface and demonstrate the CDW observed in this material is a consequence of Fermi wave vector nesting. Both the suppression of DOS at the Fermi level as well as broad spectral

features with notable cusps are in agreement with 1D electron dynamics. While the low-energy spectra are described by TLL, the dispersion of the cusps in the full energy versus momentum space in high-energy range could be only accurately reproduced by a 1D Hubbard model with suitable finite range interactions. Consequently, the cusps could be interpreted as spin- and charge- separation in these 1D metals. The accurate description of the experiment by RPDT calculations allows us to go beyond the low energy restriction of TLL, showing that the exotic 1D physics is valid for both low- and high-energy, with non-linear band dispersions and broad momentum values. Unlike other systems that only exhibit strong 1D anisotropy, the intrinsic line defects in TMDs have no specific repetition length and can thus be viewed as true 1D structures. Moreover, isolated twin grain boundaries of micrometre length have been recently reported in CVD-grown TMDs[31], which can be envisaged as remarkable candidates for quantum transport measurements on isolated 1D metals. Furthermore, 2D materials can be gated and this will exert control of transport properties of these quantum wires.

## Methods

**Sample preparation.** Monolayer $MoSe_2$ islands were grown by van der Waals epitaxy by co-deposition of atomic Se from a hot wall Se-cracker source and Mo from a mini-e-beam evaporator. The $MoS_2$ single crystal substrate was a synthetically grown and cleaved in air before introducing into the UHV chamber where it was outgassed at 300 °C for 4 h before $MoSe_2$ growth. Mo has been deposited in a selenium rich environment at a substrate temperature of ∼300–350 °C. The $MoSe_2$ monolayer was grown slowly with a growth rate of ∼0.16 monolayers per hour. While the detailed mechanism for the formation of MTBs during MBE growth is not completely understood, it has been noted that the structure shown in Fig. 1a is deficient in chalcogen atoms, i.e. the grain boundary has a stoichiometry of MoSe embedded in the $MoSe_2$ matrix. Computational studies have shown that MTBs are thermodynamically favoured over the formation of high density of individual chalcogen vacancies[15] and this may explain their presence in MBE grown samples. These samples were investigated by RT STM in a surface analysis chamber connected to the growth chamber. In Addition, characterization by VT-STM and ARPES were performed by transferring the grown samples in a vacuum suitcase to the appropriate characterization chambers. In addition, air-exposed samples were characterised by ARPES. After vacuum

annealing to ∼300 °C, the ARPES results were indistinguishable to the in vacuum transferred samples indicating the stability of the material in air against oxidation and other degradation. The stability of the sample also enables the four-point transport measurements described below.

**ARPES measurements.** Micro-ARPES measurements were performed at the ANTARES beamline at the SOLEIL synchrotron. The beam spot size was ∼120 μm. The angular and energy resolution of the beamline at a photon energy of 40 eV are ∼0.2° and ∼10 meV, respectively. Most of the data were collected around the Γ-point of the second Brillouin zone, corresponding to an emission angle of 42.5° with respect to the surface normal, for photon energy of 40 eV. Both left and right circular polarized light, as well as linear polarized light was used. The photon-incident angle on the sample was normal incidence. For circular polarized light photoemission from all MTBs is obtained. Emission from a single MTB direction could be enhanced with linear polarized light and the **A**-vector parallel to the surface. For azimuth rotation with the **A**-vector aligned to the direction of one MTB enhanced emission from this direction was obtained as shown in Fig. 3c. All data shown here were obtained at 300 K.

**Broadening of the ARPES spectral function and lifetime analysis.** As it has already been reported in previous ARPES studies (see for instance Fig. 5 of ref. 17), the lifetime of a Fermi-liquid quasi-particle, $\tau(k)$, can be directly determined from the width of the peak in the energy distribution curves (EDC), analysing the ARPES data defined by the spectral weight at fixed $k$ as a function of $\omega$, where $\omega$ is the energy. Specifically,

$$1/\tau(k) = \Delta\omega. \tag{1}$$

The consistency of a Fermi-liquid picture can be also checked by studying the momentum distribution curves (MDC), that is, from the momentum width $\Delta k$ of the spectral function peak at fixed binding energy, $\omega$. As long as the Fermi-liquid quasi-particle excitation is well defined, (that is, the decay rate is small compared with the binding energy), the energy bandwidth and momentum width are related as,

$$\Delta\omega = v_F\Delta k. \tag{2}$$

Here $v_F$ is the renormalized Fermi velocity, which can be directly measured using high energy and momentum resolution ARPES. Because of the separation of charge and spin, one hole (or one electron) is always unstable to decay into two or more elementary excitations, of which one or more carries its spin and one or more carries its charge. Then elementary kinematics implies that, at $T = 0$, the spectral function is nonzero only for negative frequencies such that,

$$|\omega| \leq \min(v_c, v_s)|k|, \tag{3}$$

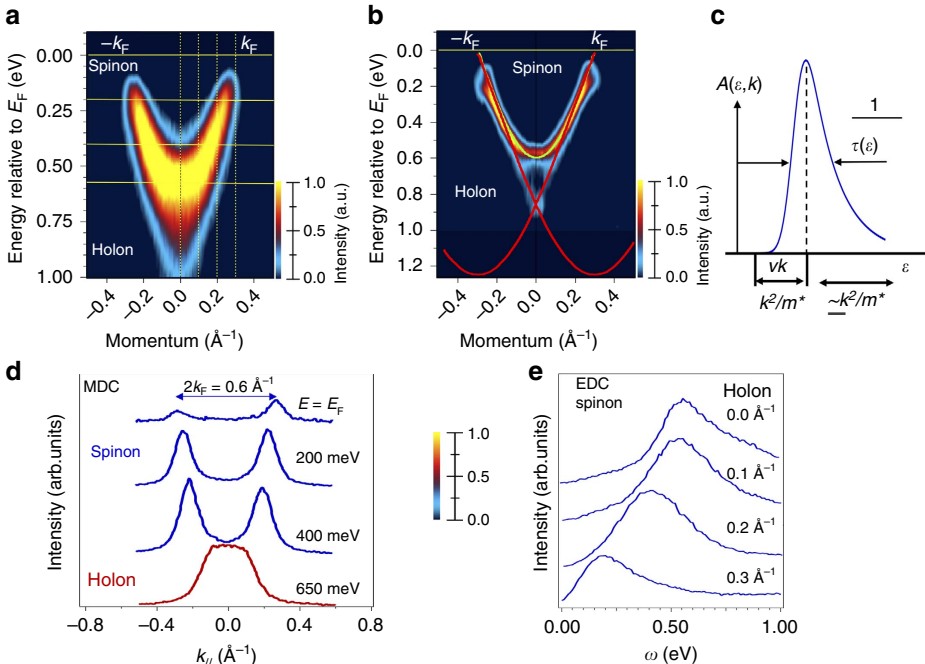

**Figure 6 | ARPES analysis using EDC and MDC plots.** (**a**) Raw ARPES data, (**b**) second derivative of data in panel (**a**), (**c**) schematic description between the EDC shape and the lifetime, (**d**) MDC plots at different binding energies extracted from panel (**a**) data and finally panel (**e**) shows EDC plots at different momentum ($k$) values indicated in panel (**a**) as yellow straight lines.

where $v_c$ and $v_s$ are the charge and spin velocity, respectively. This analysis procedure is described in Fig. 6, where the spectral function particularly at $\omega$ values between 0.40 and 0.95 eV shows a continuum, which is valid for all momentum $k$ values that fit Equation (3). MDC and EDC plots are sensitive to this detachment of the system with respect to a conventional Fermi-liquid quasi-particle behaviour.

This type of analysis, based on the shape of EDC and MDC plots, is also well explained by Emery *et al.* (see Figs 2 and 3 of ref. 5). In Fig. 6 we present the results of a similar analysis. As it is shown in panels (d) and (e), the MDC and EDC cuts of the raw data at different binding energies and momentum, respectively, show a clear enlargement of the lifetime that can be extracted from the ARPES data. However, this experimental value is just proportional to various interaction strengths. This approximative methodology of the nature and magnitude of the present interactions can be improved by using more sophisticate theoretical approaches as the one reported in the present manuscript.

**PDT as starting point of our theoretical method.** The method used in our theoretical analysis of the spin-charge separation observed in the 1D quantum-line defects of MoSe₂ was conceived for that specific goal. It combines the pseudo-fermion dynamical theory (PDT) for the 1D Hubbard model[24,27,37] with a suitable renormalization procedure.

On the one hand, the 1D Hubbard model range $\alpha_0 \in [0, 1/8]$ corresponds to the intervals $K_c \in [1/2, 1]$ and $\xi_c \in [1, \sqrt{2}]$ of the TLL charge parameter[29,12,43] and the related parameter $\xi_c = \sqrt{2K_c}$. On the other hand, the range $\alpha \in [0.75, 0.78]$ for which the renormalized theory is found to agree with the experiments implies that $\tilde{K}_c = 1 + 2\alpha - 2\sqrt{\alpha(1+\alpha)}$ and $\tilde{\xi}_c = \sqrt{2\tilde{K}_c}$ have values in the ranges $\tilde{K}_c \in [0.20, 0.21]$ and $\tilde{\xi}_c \in [0.63, 0.65]$, respectively. Here $\tilde{K}_c$ and $\tilde{\xi}_c$ is our notation for the TTL charge parameter and related parameter, respectively, in the general case when they may have values within the extended intervals $\tilde{K}_c \in [1/8, 1]$ and thus $\tilde{\xi}_c \in [1/2, \sqrt{2}]$. The minimum values $\tilde{K}_c = 1/8$ and $\tilde{\xi}_c = 1/2$ follow from corresponding phase-shift allowed ranges. (Below the relation of $\tilde{\xi}_c$ to phase shifts is reported.) The above experimental subinterval $\tilde{K}_c \in [0.20, 0.21]$ belongs to the interval $\tilde{K}_c \in [1/8, 1/2]$ for which the electron finite-range interactions must be accounted for ref. 29.

In the case of the conventional 1D Hubbard model, the PDT was the first approach to compute the spectral functions for finite values of $U/t$ near singular lines at high-energy scales beyond the low-energy TLL limit[24]. (In the low-energy limit the PDT recovers the TLL physics[37].) After the PDT was introduced for that integrable model, novel methods that rely on a mobile impurity model (MIM) approach have been developed to tackle the high-energy physics of both non-integrable and integrable 1D correlated quantum problems, also beyond the low-energy TLL limit[13,28,44,45]. The relation between the PDT and MIM has been clarified for a simpler model[46], both schemes leading to exactly the same momentum dependent exponents in the spectral functions expressions. Such a relation applies as well to more complex models. For instance, studies of the 1D Hubbard model by means of the MIM[44,45] lead to exactly the same momentum, interaction and density dependence as the PDT for the exponents that control the one-electron removal spectral function near its branch lines.

For integrable models, in our case the 1D Hubbard model, there is a representation in terms of elementary objects called within the PDT $c$ and $s$ pseudofermions for which there is only zero-momentum forward-scattering at all energy scales. The $c$ and $s$ bands momentum values are associated with the 1D Hubbard model exact Bethe-ansatz solution quantum numbers. The $c$ pseudofermion and the $s$ pseudofermion annihilated under transitions from the $N$ electron ground state to the $N-1$ electron excited states refer to the usual holon and spinon, respectively[12,13,43].

That for the pseudofermions there is only zero-momentum forward-scattering at all energy scales, follows from the existence of an infinite number of conservation laws associated with the model integrability[47,48]. This means that in contrast to the model underlying electron interactions, the pseudofermions, on scattering off each other only acquire phase shifts. Hence under their scattering events there is no energy and no momentum exchange, on the contrary of the more complex underlying physical particles interactions. In the vicinity of well-defined $(k, \omega)$-plane features called branch lines, the $T = 0$ spectral functions of integrable 1D correlated models are of power-law form with negative momentum dependent exponents. Such properties apply to all integrable 1D correlated models.

**Universality behind our method renormalization procedures.** In the case of non-integrable 1D correlated models, there is no pseudofermion representation for which there is only zero-momentum forward-scattering at all energy scales. This is because of the lack of an infinite number of conservation laws. The universality found in the framework of the MIM for the spectral functions of non-integrable and integrable 1D models[13,28] refers to specific energy scales corresponding to both the low-energy TLL spectral features and energy windows near the high-energy non-TLL branch lines singularities. In the vicinity of these lines, the $T = 0$ spectral functions of non-integrable 1D correlated models are also of power-law form with negative momentum dependent exponents.

This universality means that at both these energy scales there is for such models a suitable representation in terms of pseudofermions that undergo only

zero-momentum forward-scattering events and whose phase shifts control the spectral functions behaviours. Our renormalization scheme for adding electron finite-range interactions to the 1D Hubbard model and corresponding PDT relies on this universality. Indeed, the finite-range interactions render the model non-integrable. However, in the vicinity of the branch lines singularities the spectral function remains having the same universal behaviour. Our normalization procedure can be used for any chosen $\alpha$ value in the range $\alpha \in [\alpha_0, \alpha_{max}]$. Here $\alpha_0 \in (0, 1/8)$ is the conventional 1D Hubbard model $\alpha$ value for given $U/t$ and electronic density $n$ values. For the $U/t = 0.8$ and $n = 2/3$ values found within our description of the 1D quantum-line defects of MoSe₂ it reads $\alpha_0 \approx 1.4 \times 10^{-3}$. The maximum $\alpha$ value $\alpha_{max} = 49/32 = 1.53125$ refers through the relation $\alpha = (1 - \tilde{K}_c)^2 / 4\tilde{K}_c$, and thus $\alpha = (2 - \tilde{\xi}_c^2)^2 / 8\tilde{\xi}_c^2$ to the above minimum values $\tilde{K}_c = 1/8$ and $\tilde{\xi}_c = 1/2$.

The renormalization of the conventional 1D Hubbard model used in our studies refers to some 1D Hamiltonian with the same terms as that model plus finite-range interaction terms. The latter terms are neither a mere first-neighbouring $V$ term nor a complete long-range Coulomb potential extending over all lattice sites. Interestingly, the specific form of the additional finite-range interaction Hamiltonian terms is not needed for our study. This follows from the above universality implying that both for the low-energy TLL limit and energy windows near the high-energy branch lines singularities of the 1D Hubbard model with finite–range interactions under consideration the relation of $\alpha$ to the phase shifts remains exactly the same as for the conventional 1D Hubbard model.

Importantly, the only input parameters of our renormalization procedure are the effective $U$ and transfer integral $t$ values for which the theoretical branch lines energy bandwidths match the corresponding experimental bandwidths. Apart from the 1D quantum-line defects band-filling $n = 2/3$, our approach has no additional 'fitting parameters'.

**The spectra in terms of pseudofermion energy dispersions.** Within the PDT for the 1D Hubbard model[24–27], nearly the whole electron removal spectral weight is in the metallic phase originated by two $\iota = \pm 1$ excitations generated from the ground state by removal of one $c$ pseudofermion of momentum $q \in [-2k_F, 2k_F]$ and one $s$ pseudofermion of momentum $q' \in [-k_F, k_F]$. The superposition in the $(k, \omega)$-plane of the spectral weights associated with the corresponding two $\iota = \pm 1$ spectra generates the multi-particle continuum. Such $\iota = \pm 1$ spectra are of the form,

$$\begin{aligned} \omega(k) &= \varepsilon_c(q) + \varepsilon_s(q') \leq 0 \\ k &= -\iota 2k_F - q - q', \quad \iota = \pm 1. \end{aligned} \tag{4}$$

They are two-parametric, as they depend on the two independent $c$ and $s$ bands momenta $q$ and $q'$, respectively. Hence such spectra refer to two-dimensional domains in the $(k, \omega)$-plane. They involve the energy dispersion $\varepsilon_c(q)$ whose $c$ momentum band interval is $q \in [-\pi, \pi]$ and whose ground-state $c$ pseudofermion occupancy is $q \in [-2k_F, 2k_F]$ and the dispersion $\varepsilon_s(q')$ whose $s$ momentum band range is $q' \in [-k_F, k_F]$, which is full in the present zero spin-density ground state, are defined below.

The multi-particle continuum in the one-electron removal spectral function that results from the superposition of the spectral weights associated with the two $\iota = \pm 1$ spectra display three branch lines that display the cusps: two $c, \iota$ branch lines and a $s$ branch line. The $c, \iota$ branch lines result from processes for which the removed $c$ pseudofermion has momentum in the range $q \in [-2k_F, 2k_F]$ and the removed s pseudofermion has momentum $q' = -\iota k_F = \mp k_F$. Hence the excitation physical momentum is $k = -\iota k_F - q = \mp k_F - q$. The $s$ branch line results from removal of one $c$ pseudofermion of momentum $q = -\iota 2k_F = \mp 2k_F$. The removed $s$ pseudofermion has momentum in the interval $q' \in [-k_F, k_F]$. The physical momentum is then given by $k = -q'$.

It is convenient to redefine the two $c, \iota$ branch lines in terms of related $c$ and $c'$ branch lines. The spectra of the $c$, $c'$, and $s$ branch lines are plotted in Fig. 5d–f for $U/t = 0.8$, $t = 0.58$ eV and electronic density $n = 2/3$. On the one hand, the $c$ branch line results from processes relative to the ground state that involve removal of one $c$ pseudofermion with momentum belonging to the ranges $q \in [-2k_F, -k_F]$ and $q \in [k_F, 2k_F]$ and removal of one $s$ pseudofermion with momentum $q' = -\iota k_F$ for $\iota = \text{sgn}\{k\}$. The $c$ branch line spectrum then reads,

$$\begin{aligned} \omega_c(k) &= \varepsilon_c(|k| + k_F) \\ k &= -\text{sgn}\{k\}k_F - q \in [-k_F, k_F]. \end{aligned} \tag{5}$$

On the other hand, the $c'$ branch line is generated by removal of one $c$ pseudofermion with momentum belonging to the ranges $q \in [-2k_F, k_F]$ and $q \in [-k_F, 2k_F]$ and removal of one $s$ pseudofermion with momentum $q' = -\iota k_F$ for $\iota = -\text{sgn}\{k\}$. Its spectrum is given by,

$$\begin{aligned} \omega_{c'}(k) &= \varepsilon_c(|k| - k_F) \\ k &= \text{sgn}\{k\}k_F - q \in [-3k_F, 3k_F]. \end{aligned} \tag{6}$$

The $s$ branch line spectrum reads,

$$\begin{aligned} \omega_s(k) &= \varepsilon_s(k) \\ k &= -q' \in [-k_F, k_F]. \end{aligned} \tag{7}$$

The dispersions $\varepsilon_c(q)$ and $\varepsilon_s(q')$ appearing in these equations are uniquely defined by the following equations valid for $U/t > 0$ and electronic densities $n \in [0, 1]$,

$$
\begin{aligned}
\varepsilon_c(q) &= \bar{\varepsilon}_c(k(q)) \quad \text{for} \quad q \in [-\pi, \pi] \\
\varepsilon_s(q') &= \bar{\varepsilon}_s(\Lambda(q')) \quad \text{for} \quad q' \in [-k_F, k_F], \\
\bar{\varepsilon}_c(k) &= \int_Q^k dk' \, 2t\eta_c(k') \quad \text{for} \quad k \in [-\pi, \pi] \\
\bar{\varepsilon}_s(\Lambda) &= \int_\infty^\Lambda d\Lambda' \, 2t\eta_s(\Lambda') \quad \text{for} \quad \Lambda \in [-\infty, \infty].
\end{aligned}
\tag{8}
$$

Here the distributions $2t\eta_c(\Lambda)$ and $2t\eta_s(\Lambda)$ are the unique solutions of coupled integral equations given in supplementary Equations 1 and 2.

The $q$ and $q'$ dependence of the dispersions $\varepsilon_c(q)$ and $\varepsilon_s(q')$ occurs through that of the momentum rapidity function $k = k(q)$ for $q \in [-\pi, \pi]$ and spin rapidity function $\Lambda = \Lambda(q')$ for $q' \in [-k_F, k_F]$, respectively. Those are defined in terms of their inverse functions $q = q(k)$ for $k \in [-\pi, \pi]$ and $q' = q'(\Lambda)$ for $\Lambda \in [-\infty, \infty]$ in supplementary Equations 3 and 4. The distributions $2\pi\rho(k)$ and $2\pi\sigma(\Lambda)$ in their expressions are the unique solutions of the coupled integral equations provided in Supplementary Equations 5 and 6.

**Spectral function within the conventional 1D Hubbard model.** Within the PDT for the 1D Hubbard model[24–27], the spectral weight distributions are controlled by the set of phase shifts $\pm 2\pi\Phi_{\beta,\beta'}(q, q')$ acquired by the $\beta = c$ and $\beta = s$ pseudofermions with momentum $q$ upon scattering off each $\beta' = c$ and $\beta' = s$ pseudofermion with momentum $q'$ created $(+)$ or annihilated $(-)$ under the transitions from the ground state to the excited energy eigenstates. (In contrast to otherwise in this section, here the momentum values $q$ and $q'$ are not necessarily those of c and s pseudofermions, respectively.)

The expressions of the momentum dependent exponents that control the line shape in the vicinity of the $\gamma = c, c', s$ branch lines involve phase shifts whose $\beta = c, s$ pseudofermions have momentum at the corresponding Fermi points, $\pm q_{Fc} = \pm 2k_F$ and $\pm q_{Fs} = \pm k_F$. This includes phase shifts $2\pi\Phi_{\beta,\beta'}(\iota q_{F\beta}, \iota' q_{F\beta'}) = -2\pi\Phi_{\beta,\beta'}(-\iota q_{F\beta}, -\iota' q_{F\beta'})$, where $\iota = \pm 1$, $\iota' = \pm 1$, acquired by such $\beta = c, s$ pseudofermions on scattering off $\beta' = c, s$ pseudofermions of momentum also at Fermi points annihilated under the transitions from the $N$ electron ground state to the $N - 1$ excited states. Furthermore, such exponents expressions also involve phase shifts $-2\pi\Phi_{\beta,c}(q_{F\beta}, q) = 2\pi\Phi_{\beta,c}(-q_{F\beta}, -q)$ and $-2\pi\Phi_{\beta,s}(q_{F\beta}, q') = 2\pi\Phi_{\beta,s}(-q_{F\beta}, -q')$ acquired by the same $\beta = c, s$ pseudofermions upon scattering off $\beta' = c$ and $\beta' = s$ pseudofermions of momentum $q \in [-2k_F, 2k_F]$ and $q' \in [-k_F, k_F]$, respectively, annihilated under such transitions.

For energy windows corresponding to small energy deviations $(\omega_\gamma(k) - \omega) > 0$ from the high-energy $\gamma = c, c', s$ branch-line spectra $\omega_c(k) = \varepsilon_c(|k| + k_F)$ for $k \in (-k_F, k_F)$, $\omega_{c'}(k) = \varepsilon_c(|k| - k_F)$ for $k \in (-3k_F, 3k_F)$ and $\omega_s(k) = \varepsilon_s(k)$ for $k \in (-k_F, k_F)$, equations 5–7, the electron removal spectral function has within the PDT the universal form[25–27,37],

$$
B(k, \omega) \propto (\omega_\gamma(k) - \omega)^{\zeta_\gamma(k)} \quad \text{for} \quad \gamma = c, c', s.
\tag{9}
$$

The exponents in this general expression are for $U/t > 0$ and electronic densities $n \in [0, 1]$ given in terms of pseudofermion phase shifts in units of $2\pi$ by,

$$
\begin{aligned}
\zeta_c(k) &= -\frac{1}{2} + \sum_{\iota = \pm 1} \left( \frac{\xi_c}{4} + \text{sgn}\{k\}\Phi_{c,c}(\iota 2k_F, q) \right)^2 \\
& k = \in [-k_F, k_F], \\
& q = -\text{sgn}\{k\}k_F - k \in [-2k_F, -k_F]; [k_F, 2k_F], \\
\zeta_{c'}(k) &= -\frac{1}{2} + \sum_{\iota = \pm 1} \left( \frac{\xi_c}{4} - \text{sgn}\{k\}\Phi_{c,c}(\iota 2k_F, q) \right)^2 \\
& k = \in [-3k_F, 3k_F], \\
& q = \text{sgn}\{k\}k_F - k \in [-2k_F, k_F]; [-k_F, 2k_F]. \\
\zeta_s(k) &= -1 + \sum_{\iota = \pm 1} \left( \frac{\iota}{2\xi_c} + \Phi_{c,s}(\iota 2k_F, q') \right)^2 \\
& k \in [-k_F, k_F] \quad \text{and} \quad q' = -k \in [-k_F, k_F].
\end{aligned}
\tag{10}
$$

At zero spin density, the entries of the conformal-field theory dressed-charge matrix $Z$ and corresponding matrix $(Z^{-1})^T$ can be alternatively expressed in terms of pseudofermion phase shifts in units of $2\pi$ and of the related parameters $\xi_c$ and $\xi_s$, as given supplementary equations 7 and 8, respectively. (Here we use the dressed-charge matrix definition of ref. 37, which is the transposition of that of ref. 43.) Conversely, the pseudofermion phase shifts with both momenta at the Fermi points can be expressed in terms of only the charge TLL parameter $K_c = \xi_c^2/2$ and spin TLL parameter $K_s = \xi_s^2/2$ (ref. 43) and thus of the present

related $\beta = c, s$ parameters $\xi_\beta = \sqrt{2K_\beta}$. Specifically,

$$
\begin{aligned}
2\pi\Phi_{\beta,\beta'}(\iota q_{F\beta}, q_{F\beta'}) &= \iota 2\pi\Phi_{\beta,\beta'}(q_{F\beta}, \iota q_{F\beta'}) \\
&= \frac{\pi(\xi_\beta - 1)^2}{\xi_\beta} \quad \text{for} \quad \beta = \beta', \quad \iota = +1, \\
&= -\frac{\pi(\xi_\beta^2 - 1)}{\xi_\beta} \quad \text{for} \quad \beta = \beta', \quad \iota = -1, \\
&= (-\iota)^{\delta_{\beta,s}} \frac{\pi}{2} \xi_\beta \quad \text{for} \quad \beta \neq \beta', \quad \iota = \pm 1.
\end{aligned}
\tag{11}
$$

Here $\beta = c, s$ and $\beta' = c, s$.

The two sets of two coupled integral equations, Supplementary equations 1, 2, 5 and 6, respectively, that one must solve to reach the momentum dependence of the exponents, equation 10, have no simple analytical solution. Within our study, these equations are solved by exact numerical methods. The exponents found from such a numerical solution are plotted as a function of the momentum $k$ in Fig. 5a–c (dashed-dotted lines) for $U/t = 0.8$, $t = 0.58$ eV and electronic density $n = 2/3$. The c and s exponent expressions in Equation 10 are not valid at the low-energy limiting values $k = \pm k_F$.

In the present zero spin-density case, the spin $SU(2)$ symmetry implies that the parameter $\xi_s$ appearing in Equation 11 is $u$ independent and reads $\xi_s = \sqrt{2}$. The parameter $\xi_c$ in Equations 10 and 11 is in turn given by $\xi_c = f(\sin Q/u)$ where the function $f(r)$ is the unique solution of the integral equation given the Supplementary Equation 9 whose kernel $D(r)$ is defined in Supplementary Equation 10. The parameter $\xi_c \in [1, \sqrt{2}]$ has limiting values $\xi_c = \sqrt{2}$ for $u \to 0$ and $\xi_c = 1$ for $u \to \infty$. This is why for the 1D Hubbard model the exponent in the low-$\omega$ power law dependence of the electronic density of states suppression $|\omega|^{\alpha_0}$,

$$
\alpha_0 = \frac{(1 - K_c)^2}{4K_c} = \frac{(2 - \xi_c^2)^2}{8\xi_c^2} \in [0, 1/8],
\tag{12}
$$

has corresponding limiting values $\alpha_0 = 0$ for $u \to 0$ and $\alpha_0 = 1/8$ for $u \to \infty$.

The c pseudofermion phase shifts $2\pi\Phi_{c,c}(\iota 2k_F, q)$ for $q \in [-2k_F, 2k_F]$ and $2\pi\Phi_{c,s}(\iota 2k_F, q')$ for $q' \in [-k_F, k_F]$ that determine the momentum dependence of the exponents in equation 10 are beyond the reach of the TTL. Such exponents also involve the s pseudofermion phase shifts $2\pi\Phi_{s,c}(\iota k_F, q)$ and $2\pi\Phi_{s,s}(\iota k_F, q')$. Because of the spin $SU(2)$ symmetry, at zero spin density the latter phase shifts are $u$ independent. They are given in the supplementary Equations 14 and 15. Their values provided in these equations have been accounted for in the derivation of the exponents expressions in Equation (10) and contribute to them.

The c pseudofermion phase shifts explicitly appearing in the exponents expressions, Equation (10), can be written as $2\pi\Phi_{c,c}(\iota 2k_F, q) = 2\pi\bar{\Phi}_{c,c}(\iota \sin Q/u, \sin k(q)/u)$ and $2\pi\Phi_{c,s}(\iota 2k_F, q') = 2\pi\bar{\Phi}_{c,s}(\iota \sin Q/u, \Lambda(q')/u)$ where the parameters $\pm Q = k(\pm 2k_F)$ define the c pseudofermion Fermi points in rapidity space. The corresponding general c pseudofermion phase shifts are given by $2\pi\Phi_{c,c}(q, q') = 2\pi\bar{\Phi}_{c,c}(\sin k(q)/u, \sin k(q')/u)$ and $2\pi\Phi_{c,s}(q, q') = 2\pi\bar{\Phi}_{c,s}(\sin k(q)/u, \Lambda(q')/u)$ where the related rapidity phase shifts $2\pi\bar{\Phi}_{c,c}(r, r')$ and $2\pi\bar{\Phi}_{c,s}(r, r')$ are the unique solutions of the integral equations given in the Supplementary Equations 11 and 12. The free term $D_0(r)$ of the former integral equation is provided in Supplementary Equation 13.

One finds from manipulations of integral equations that the energy dispersions $\varepsilon_c(q)$ and $\varepsilon_s(q)$, equation (8), can be expressed exactly in terms of the c pseudofermion rapidity phase shifts as follows,

$$
\begin{aligned}
\varepsilon_c(q) &= \varepsilon_c^0(q) - \varepsilon_c^0(2k_F), \\
\varepsilon_c^0(q) &= -2t\cos k(q) \\
&+ \frac{t}{\pi} \int_{-Q}^Q dk \, 2\pi\bar{\Phi}_{c,c}\left( \frac{\sin k}{u}, \frac{\sin k(q)}{u} \right) \sin k,
\end{aligned}
\tag{13}
$$

and

$$
\begin{aligned}
\varepsilon_s(q') &= \varepsilon_s^0(q') - \varepsilon_s^0(k_F), \\
\varepsilon_s^0(q') &= \frac{t}{\pi} \int_{-Q}^Q dk \, 2\pi\bar{\Phi}_{c,s}\left( \frac{\sin k}{u}, \frac{\Lambda(q')}{u} \right) \sin k,
\end{aligned}
\tag{14}
$$

respectively. Here $k = k(q)$ and $\Lambda = \Lambda(q')$ are the momentum rapidity function and spin rapidity function, respectively, considered above.

**Description of the finite-range interactions within our method.** Below it is confirmed that except for the effective $U$ value the energy dispersions, equations (13) and (14), are not affected by the renormalization that accounts for the short–range interactions. As reported above, the effective value $U = 0.8t$ is determined by the ratio $W_h/W_s$ of the experimentally observed c band (holon) and s band (spinon) energy bandwidths $W_h = \varepsilon_c(2k_F) - \varepsilon_c(0)$ and $W_s = \varepsilon_s(k_F) - \varepsilon_c(0)$, respectively. Indeed, within the 1D Hubbard model the $W_h/W_s$ value only depends on $U/t$ and the electronic density $n$. For $n = 2/3$ the agreement with the observed energy bandwidths is then found to be reached for $U/t = 0.8$.

However, the renormalization fixes the effective $U$ value yet does not affect $t$. This is because of symmetry implying that within the 1D Hubbard model the full $c$ band energy bandwidth $\varepsilon_c(\pi) - \varepsilon_c(0)$ is independent of $U$ and $n$ and exactly reads $4t$. That energy bandwidth can be written as $W_h + W_c = 4t$, where for the present metallic phase the energy bandwidth $W_c = \varepsilon_c(\pi) - \varepsilon_c(2k_F)$ is finite. Within our pseudofermion representation, $W_h$ and $W_c$ are the $c$ band filled and unfilled, respectively, ground-state Fermi sea energy bandwidths. Again, the value of the ratio $W_h/W_c$ only depends on $U/t$ and the electronic density $n$. Accounting for the $W_h/W_c$ value at $U/t = 0.8$ and $n = 2/3$ together with the exact relation $W_h + W_c = 4t$ one finds from analysis of Fig. 5d–f that $t \approx 0.58$ eV for the MoSe$_2$ 1D quantum-line defects.

Such defects experimental uncertainty interval $\alpha \in [0.75, 0.80]$ of the exponent that controls the low-$\omega$ electronic density of states suppression $|\omega|^\alpha$ is outside the corresponding 1D Hubbard model range, Equation (12). Hence the $U = 0.8t$ value obtained from matching the corresponding ARPES cusps lines spectra with those of the 1D Hubbard model for electronic density $n = 2/3$ refers to an effective interaction having contributions both from electron onsite and finite-range interactions. In addition to the interaction $U$ renormalization, both the parameter $\xi_c$ and the corresponding $c$ pseudofermion phase shifts $2\pi\Phi_{c,\beta'}(i2k_F, q_{F\beta'})$ in equation (11), where $\beta' = c,s$ whose expressions involve $\xi_c$ undergo a second renormalization. It is such that $\xi_c$ is replaced by a parameter $\tilde{\xi}_c$ associated with $\alpha$ values in the range $\alpha \in (\alpha_0, \alpha_{max})$.

The universality referring to low-energy values in the vicinity of the $c$ and $s$ bands Fermi points implies that for the non-integrable model with finite-range interactions the relation $\alpha_0 = (2 - \xi_c^2)^2 / 8\xi_c^2$ given in Equation (12) remains having the same form for $\alpha \in [\alpha_0, \alpha_{max}]$ and $\tilde{\xi}_c \in [1/2, \xi_c]$, so that,

$$\alpha = \frac{(2 - \tilde{\xi}_c^2)^2}{8\tilde{\xi}_c^2}; \quad \tilde{\xi}_c = \sqrt{2\left(1 + 2\alpha - 2\sqrt{\alpha(1+\alpha)}\right)}. \quad (15)$$

(The first equation other mathematical solution, $\tilde{\xi}_c = \sqrt{2(1 + 2\alpha + 2\sqrt{\alpha(1+\alpha)})}$, is not physically acceptable.)

On the one hand, the spin $SU(2)$ symmetry imposes that the values of the $U/t$-independent parameter $\xi_s = \sqrt{2}$ and $s$ pseudofermion phase shifts $2\pi\Phi_{s,\beta'}(i k_F, q_{F\beta'})$ in equation (11) where $\beta' = c, s$ remain unchanged for the model with finite-range interactions. On the other hand, the general relations, equation (11), are universal so that for that model corresponding to any $\alpha$ value in the range $\alpha \in [\alpha_0, \alpha_{max}]$ the $c$ pseudofermion phase shifts $2\pi\Phi_{c,\beta'}(i2k_F, q_{F\beta'})$ are for $\beta' = c, s$ given by,

$$2\pi\tilde{\Phi}_{c,c}(i2k_F, 2k_F) = i2\pi\tilde{\Phi}_{c,c}(2k_F, i2k_F)$$
$$= \frac{\pi(\tilde{\xi}_c - 1)^2}{\tilde{\xi}_c} \quad \text{for} \quad i = +1,$$
$$= \frac{\pi(\tilde{\xi}_c^2 - 1)}{\tilde{\xi}_c} \quad \text{for} \quad i = -1, \quad (16)$$
$$2\pi\tilde{\Phi}_{c,s}(i2k_F, k_F) = i2\pi\tilde{\Phi}_{c,s}(2k_F, i k_F)$$
$$= \frac{\pi}{2}\tilde{\xi}_c \quad \text{for} \quad i = \pm 1.$$

The universality on which our scheme relies refers both to the low-energy TLL limit and to energy windows near the high-energy $c$, $c'$ and $s$ branch-lines singularities. The expression of the exponents that control the spectral function behaviour at low energy and in the vicinity of such singularities only involves the phase shifts of $c$ and $s$ pseudofermions with momenta at their Fermi points $q = \pm 2k_F$ and $q' = \pm k_F$, respectively. On the one hand, as result in part of the spin $SU(2)$ symmetry, at zero spin density the general $s$ pseudofermion phase shifts $2\pi\tilde{\Phi}_{s,s}(q', q)$ and $2\pi\tilde{\Phi}_{s,c}(q', q)$ remain unchanged for their whole momentum intervals. On the other hand, the general phase shifts $2\pi\tilde{\Phi}_{c,c}(q, q')$ and $2\pi\tilde{\Phi}_{c,s}(q, q')$ of $c$ pseudofermions whose momenta have absolute values $|q| < 2k_F$ inside the $c$ band Fermi sea contribute neither to the TLL low-energy spectral function expression nor to the high-energy branch-lines exponents. Consistently, similarly to the $s$ pseudofermion phase shifts $2\pi\tilde{\Phi}_{s,s}(q', q)$ and $2\pi\tilde{\Phi}_{s,c}(q', q)$ they remain unchanged upon increasing $\alpha$ from $\alpha = \alpha_0$.

Hence the main issue here is the renormalization of phase shifts of $c$ pseudofermions with momenta at the Fermi points, $2\pi\tilde{\Phi}_{c,c}(i2k_F, q)$ and $2\pi\tilde{\Phi}_{c,s}(i2k_F, q')$ for $i = \pm 1$. Multiplying $2\pi\tilde{\Phi}_{c,c}(i2k_F, q)$ and $2\pi\tilde{\Phi}_{c,s}(i2k_F, q')$ by the phase factor $-1$ gives the phase shifts acquired by the $c$ pseudofermions of momenta $q = i2k_F = \pm 2k_F$ on scattering off one $c$ band hole (holon) created under a transition to an excited state at any momentum $q$ in the interval $q \in [-2k_F, 2k_F]$ and one $s$ band hole (spinon) created at any momentum $q'$ in the domain $q' \in (-k_F, k_F)$, respectively. The overall phase-shift renormalization must preserve the $c$ pseudofermion phase-shifts values given in Equation (16) for (i) $q = i2k_F = \pm 2k_F$ and (ii) $q' = i k_F = \pm k_F$. Hence it introduces suitable factors multiplying $2\pi\tilde{\Phi}_{c,c}(i2k_F, q)$ and $2\pi\tilde{\Phi}_{c,s}(i2k_F, q')$. In the case of $2\pi\tilde{\Phi}_{c,c}(i2k_F, q)$, this brings about a singular behaviour at $q = -i2k_F$ for $\alpha > \alpha_0$ similar to that in the $s$ pseudofermion phase shift $2\pi\tilde{\Phi}_{s,s}(i k_F, q')$ at $q' = i k_F$, Supplementary equation 15, for the conventional 1D Hubbard model, which remains having the same values for the renormalized model.

The $c$ and $s$ pseudofermion phase shifts of the 1D Hubbard model with electron finite-range interactions are for the whole range $\alpha \in [\alpha_0, \alpha_{max}]$ thus of the general form,

$$2\pi\tilde{\Phi}_{c,c}(q, q') = 2\pi\Phi_{c,c}(q, q') \quad \text{for} \quad q \neq i2k_F, \quad i = \pm 1,$$
$$2\pi\tilde{\Phi}_{c,c}(i2k_F, q') = \frac{\xi_c(\tilde{\xi}_c - 1)(\tilde{\xi}_c - (-1)^{\delta_{q',-i2k_F}})}{\tilde{\xi}_c(\xi_c - 1)(\xi_c - (-1)^{\delta_{q',-i2k_F}})}$$
$$\times 2\pi\Phi_{c,c}(i2k_F, q) \quad \text{for} \quad i = \pm 1,$$
$$2\pi\tilde{\Phi}_{c,s}(q, q') = 2\pi\Phi_{c,s}(q, q') \quad \text{for} \quad q \neq i2k_F, \quad i = \pm 1, \quad (17)$$
$$2\pi\tilde{\Phi}_{c,s}(i2k_F, q') = \frac{\tilde{\xi}_c}{\xi_c} 2\pi\Phi_{c,s}(i2k_F, q') \quad \text{for} \quad i = \pm 1,$$
$$2\pi\tilde{\Phi}_{s,s}(q', q) = 2\pi\Phi_{s,s}(q', q),$$
$$2\pi\tilde{\Phi}_{s,c}(q', q) = 2\pi\Phi_{s,c}(q', q).$$

Our theoretical results refer to the thermodynamic limit at $T = 0$. In that case the phase-shifts renormalization, Equation (17), only affects those of the $c$ pseudofermion scatterers with momentum values $\pm 2k_F$ corresponding to the zero-energy Fermi level. Note however that the corresponding $c$ and $s$ pseudofermion scattering centres have momenta $q \in [-2k_F, 2k_F]$ and $q' \in [-k_F, k_F]$, respectively, that correspond to a large range of high-energy values. At finite temperature $T \approx 300$ K one has that $k_B T \approx 0.045t$ where $t \approx 0.58$ eV is within the present theoretical description the transfer integral value suitable for the MoSe$_2$ 1D quantum-line defects. The derivation of some of the theoretical expressions involves a $T = 0$ $c$ band momentum distribution that reads one for $|q| < 2k_F$ and zero for $2k_F < |q| < \pi$. At finite temperature $T \approx 300$ K, such a distribution is replaced by a $c$ pseudofermion Fermi-Dirac distribution. This implies for instance that the $q = \pm 2k_F$ $c$ pseudofermion phase-shift renormalization in Equation (17) is extended from the zero-energy Fermi level to a small region of energy bandwidth $0.045t \approx 0.026$ eV near the $c$ band Fermi points $q = \pm 2k_F$. This refers to a corresponding small region with the same energy bandwidth near the physical Fermi points $k = \pm k_F$ in Fig. 5d–f. Interestingly, finite-size effects have at $T = 0$ the similar effect of slightly enhancing the energy bandwidth of the $c$ pseudofermion phase shifts renormalization, Equation (17), in the very vicinity of the zero-energy Fermi level. Hence any small finite temperature and/or the system finite size remove/s the singular behaviour of the phase-shifts renormalization being restricted to the zero-energy Fermi level.

Fortunately, both the finite size of the MoSe$_2$ 1D quantum-line defects and the experimental temperature $\approx 300$ K lead though to very small effects, as confirmed by the quantitative agreement reached between the $T = 0$ theoretical results associated with the 1D Hubbard model with electron finite-range interactions and the experimental data. Hence for simplicity in the following we remain using our $T = 0$ theoretical analysis in terms of that model in the thermodynamic limit.

**Spectral function accounting for finite-range interactions.** For energy windows corresponding to small $\gamma = c, c', s$ energy deviations $(\omega_\gamma(k) - \omega) > 0$ from the high-energy branch-line spectra $\omega_\gamma(k)$ given in equations 5–7, which as confirmed below remain unchanged upon increasing $\alpha$ from $\alpha_0$, the general form of the electron removal spectral function, equation 9 and corresponding exponent, equation 10, prevails for the model with finite-range interactions corresponding to $\alpha \in [\alpha_0, \alpha_{max}]$. Hence for these energy windows that spectral function has the same universal form as in equation 9,

$$B(k, \omega) \propto (\omega_\gamma(k) - \omega)^{\tilde{\zeta}_\gamma(k)} \quad \text{for} \quad \gamma = c, c', s. \quad (18)$$

Both within the PDT ($\alpha = \alpha_0$) and RPDT ($\alpha > \alpha_0$), most of the one-electron spectral weight is located in the $(k, \omega)$-plane at and near the singular branch lines. Those refer to the $k$ ranges of the $\gamma = c, c', s$ branch lines for which the corresponding exponent $\tilde{\zeta}_\gamma(k)$ in equation 18 is negative. For further information on the validity of the spectral functions expressions, equations (9) and (18), and the definition of some quantities used in our theoretical analysis, see Supplementary note 3.

We start by confirming that the $c$ and $s$ pseudofermion energy dispersions in the expressions of the $\gamma = c, c', s$ branch-lines spectra $\omega_\gamma(k)$, equations 5–7, remain unchanged. This follows from the behaviour of the phase shifts appearing in these pseudofermion energy dispersions expressions, equations (13) and (14). In the case of the conventional 1D Hubbard model, the integral $\int^Q dk$ over the rapidity momentum $k$ in the integrand rapidity phase shifts $2\pi\bar{\Phi}_{c,c}(\sin k/u, \sin k(q)/u)$ and $2\pi\bar{\Phi}_{c,s}(\sin k/u, \Lambda(q')/u)$ of equations (13) and (14) can be transformed into a momentum integral $\int_{-2k_F}^{2k_F} dq''$ over the whole $c$ band Fermi sea with the integration momentum $q'' \in [-2k_F, 2k_F]$ appearing in corresponding integrand $c$ pseudofermion phase shifts $2\pi\Phi_{c,c}(q'', q)$ and $2\pi\Phi_{c,s}(q'', q')$, respectively.

Under the electron finite-range interactions renormalization, the latter phase shifts become $2\pi\tilde{\Phi}_{c,c}(q'', q)$ and $2\pi\tilde{\Phi}_{c,s}(q'', q')$, respectively, as defined in Equation (17). As given in that equation, the latter $c$ pseudofermion phase shifts are only renormalized at the Fermi points, $q'' = \pm 2k_F$. Hence such phase shifts renormalized values refer only to the limiting values of the integration $\int_{-2k_F}^{2k_F} dq''$. The phase-shift contributions associated with such limiting momentum values $-2k_F$ and $+2k_F$ have in the thermodynamic limit vanishing measure relative to the phase-shift contributions from the range $-2k_F < q'' < 2k_F$ in $\int_{-2k_F}^{2k_F} dq''$. For $|q''| < 2k_F$ the phase shifts $2\pi\tilde{\Phi}_{c,c}(q'', q)$ and $2\pi\tilde{\Phi}_{c,s}(q'', q')$ remain unchanged, see equation (17). Hence the energy dispersions

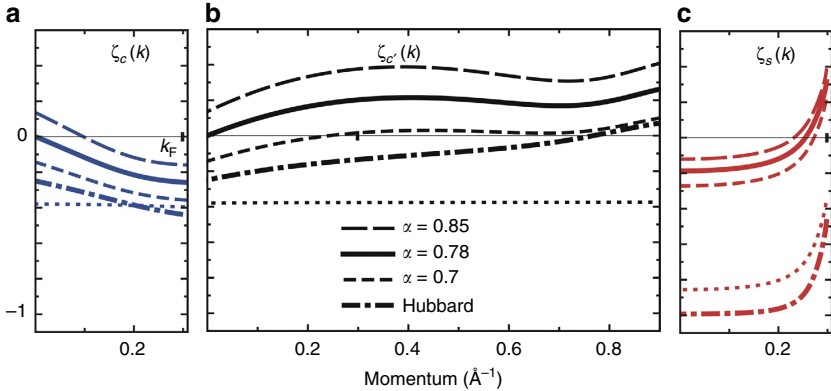

**Figure 7 | Momentum dependence of spectral-function exponents.** (**a**), (**b**) and (**c**): The c, c′ and s branch-lines exponents, respectively, defined in Equation (19) plotted as a function of the momentum $k$ for $U/t = 0.8$, $t = 0.58$ eV, $n = 2/3$ and representative $\alpha$ values $\alpha = \alpha_0 \approx 1.4 \times 10^{-3}$, $\alpha = 0.70$, $\alpha = 0.7835 \approx 0.78$ and $\alpha = 0.85$. In addition, the dotted lines refer to $\alpha = 1/8$. As justified in the text, for $\alpha \in (0.75, 0.78)$ the momentum ranges of the c, c′ and s branch lines for which such exponents are negative coincide with those showing ARPES peaks in Fig. 5e,f.

$\varepsilon_c(q) = \varepsilon_c^0(q) - \varepsilon_c^0(2k_F)$, equation (13), and $\varepsilon_s(q) = \varepsilon_s^0(q) - \varepsilon_s^0(k_F)$, equation (14), remain as well unchanged. The same thus applies to the $\gamma = c, c', s$ spectra $\omega_\gamma(k)$, equations 5–7, in the spectral function expression, equation (18).

In contrast, one finds from the combined use of equations (10) and (17) that for the model with finite–range interactions the momentum dependent exponents in that expression are renormalised. For $U/t > 0$, electronic densities $n \in [0, 1]$ and $\alpha \in [\alpha_0, \alpha_{max}]$ they are given by,

$$\tilde{\zeta}_c(k) = -\frac{1}{2} + \sum_{\iota = \pm 1} \left( \frac{\tilde{\xi}_c}{4} + \text{sgn}\{k\}\tilde{\Phi}_{c,c}(\iota 2k_F, q) \right)^2$$
$$k = \in [-k_F, k_F],$$
$$q = -\text{sgn}\{k\}k_F - k \in [-2k_F, -k_F]; [k_F, 2k_F],$$

$$\tilde{\zeta}_{c'}(k) = -\frac{1}{2} + \sum_{\iota = \pm 1} \left( \frac{\tilde{\xi}_c}{4} - \text{sgn}\{k\}\tilde{\Phi}_{c,c}(\iota 2k_F, q) \right)^2 \quad (19)$$
$$k = \in [-3k_F, 3k_F],$$
$$q = \text{sgn}\{k\}k_F - k \in [-2k_F, k_F]; [-k_F, 2k_F].$$

$$\tilde{\zeta}_s(k) = -1 + \sum_{\iota = \pm 1} \left( \frac{\iota}{2\tilde{\xi}_s} + \tilde{\Phi}_{c,s}(\iota 2k_F, q') \right)^2$$
$$k \in (-k_F, k_F) \quad \text{and} \quad q' = -k \in [-k_F, k_F].$$

Plotting the momentum dependence of these exponents requires again the use of exact numerical methods to solve the corresponding sets of coupled integral equations. The momentum dependences found from that exact numerical solution are plotted in Fig. 7 as a function of the momentum $k$ for $U/t = 0.8$, $t = 0.58$ eV, $n = 2/3$ and representative $\alpha$ values $\alpha = \alpha_0 \approx 1.4 \times 10^{-3}$, $\alpha = 0.70$, $\alpha = 0.7835 \approx 0.78$ and $\alpha = 0.85$. Their choice is confirmed below to be suitable for the discussion of the relation between the theoretical results and the observed spectral features.

The physics associated with the $\alpha$ range $\alpha \in [\alpha_0, 1/8]$ is qualitatively different from that corresponding to $\alpha \in [1/8, \alpha_{max}]$. Note that at $\alpha = 1/8$ and thus $\tilde{\xi}_c = 1$ the c pseudofermion phase shift $2\pi\Phi_{c,c}(\iota 2k_F, q)$ in equation (17) exactly vanishes. This vanishing marks the transition between the two physical regimes. The c pseudofermion phase shift $2\pi\Phi_{c,c}(\iota 2k_F, q)$ of the conventional 1D Hubbard model also vanishes in the limit of infinity onsite repulsion in which $\alpha_0 = 1/8$. Increasing $\alpha$ from $\alpha = \alpha_0$ within the interval $\alpha \in [\alpha_0, 1/8]$ indeed increases the actual onsite repulsion, which for $\alpha > \alpha_0$ is not associated anymore with the renormalised model constant effective $U$ value. In addition, it introduces electron finite-range interactions. On the one hand, in that $\alpha$ interval the effects on the $\gamma = c, c', s$ exponents, equation (19), of increasing $\alpha$ are controlled by the increase of the actual onsite repulsion. On the other hand, as $\alpha$ changes within the interval $\alpha \in [\alpha_0, 1/8]$ the fixed effective $U$ value accounts for both effects from the actual onsite interaction and emerging finite-range interactions. It imposes that the c and s pseudofermion energy dispersions in equations (13) and (14) remain as for that $U$ value. This means that the effects of increasing the actual onsite repulsion due to increasing $\alpha$ are on the matrix elements of the electron annihilation operator between energy eigenstates that control the branch-lines exponents, equation (19), and thus the spectral weights.

For $U/t = 0.8$, $t = 0.58$ eV and $n = 2/3$ the c, c′ and s branch-lines exponents, equation (19), corresponding to $\alpha = 1/8$ are represented in Fig. 7a–c, respectively, by the dotted lines. The changes in these exponents caused by increasing the $\alpha$ value from $\alpha_0$ to $1/8$ relative to the exponents curves given for the $\alpha_0 \approx 1.4 \times 10^{-3}$ conventional 1D Hubbard model in that figure are qualitatively similar to those originated by increasing $U/t$ from 0.8 to infinity within the latter model. Such an increase also enhances $\alpha_0$ from $\alpha_0 \approx 1.4 \times 10^{-3}$ to 1/8. The main difference relative

to the conventional 1D Hubbard model is that the c and s pseudofermion energy dispersions remain unchanged on increasing $\alpha$. Comparison of the momentum intervals of the $\gamma = c, c', s$ branch lines for which the exponents, Equation (19), are negative for $\alpha \in (\alpha_0, 1/8)$ with those in which there are cusps in the experimental dispersions of Fig. 5e,f reveals that there is no agreement between theory and experiments for that $\alpha$ range.

Further increasing $\alpha$ within the interval $\alpha \in [1/8, \alpha_{max}]$ corresponds to a different physics. The changes in the branch-lines exponents, equation (19), are then mainly due to the increasing effect of the electron finite–range interactions on increasing $\alpha$. It leads in general to a corresponding increase of the three $\gamma = c, c', s$ exponents $\tilde{\zeta}_\gamma(k)$, equation (19). For $U/t = 0.8$, $t = 0.58$ eV, $n = 2/3$ and both $\alpha \in [1/8, 0.75]$ and $\alpha \in [0.78, \alpha_{max}]$ the momentum intervals of the $\gamma = c, c', s$ branch lines for which these exponents are negative do not agree to those for which there are cusps in the MoSe$_2$ 1D quantum-line defects measured spectral function. To illustrate the $\alpha$ dependence of the $\gamma = c, c', s$ branch lines exponents, Equation (19), their $k$ dependence has been plotted in Fig. 7 for the set of representative $\alpha$ values $\alpha = \alpha_0 \approx 1.4 \times 10^{-3}$, $\alpha = 0.70$, $\alpha = 0.7835 \approx 0.78$ and $\alpha = 0.85$.

The following analysis refers again to the values $U/t = 0.8$, $t = 0.58$ eV and $n = 2/3$ associated with the MoSe$_2$ 1D quantum-line defects. For $\alpha < 0.75$ the momentum width of the $\gamma = c'$ branch line $k$ range for which its exponent $\tilde{\zeta}_{c'}(k)$ is negative is larger than that of the experimental dispersion shown in Fig. 5(e), (f) near the corresponding excitation energy $\approx 0.95$ meV. On increasing $\alpha$ from $\alpha = 0.75$, the $\gamma = c'$ branch line momentum width for which $\tilde{\zeta}_{c'}(k)$ is negative continuously decreases, vanishing at $\alpha = 0.7835 \approx 0.78$. Comparison of the momentum ranges for which the exponents plotted in Fig. 7 are negative with those in which there are cusps in the experimental dispersions of Fig. 5 (e) e (f) reveals that there is quantitative agreement for $\alpha \in [0.75, 0.78]$. Further increasing $\alpha$ from $\alpha = 0.78$ leads to a c branch line momentum width around $k = 0$ in which the exponent $\tilde{\zeta}_c(k)$ becomes positive. This disagrees with the observation of experimental cusps near the excitation energy $\approx 0.85$ meV around $k = 0$ and for decreasing energy along the c branch line upon further increasing $\alpha$.

That there is quantitative agreement between theory and the experiments for $\alpha \in (0.75, 0.78)$ is fully consistent with the corresponding $\alpha$ uncertainty range $\alpha \in [0.75, 0.80]$ found independently from the DOS suppression experiments. The momentum dependence of the $\gamma = c, c', s$ branch lines exponents corresponding to $\alpha = 0.78$ is represented by full lines in Fig. 5a,c and d for $U = 0.8t$, $t = 0.58$ eV and electronic density $n = 2/3$.

As for the exponents expressions, Equation (10), those of the c and s branch-line exponents given in Equation (19) are not valid at the low-energy limiting values $k = \pm k_F$. While in the thermodynamic limit this refers to $k = \pm k_F$, for the finite-size MoSe$_2$ 1D quantum-line defects it may refer to two small low-energy regions in the vicinity of $k = \pm k_F$. Both this property and the positivity of the s branch exponent for $\alpha \in (0.75, 0.78)$ in these momentum regions are consistent with the lack of low-energy cusps in the ARPES data shown in Fig. 5e,f.

We have calculated the $k$ and $\omega$ dependence of the spectral function expression of the 1D Hubbard model with finite-range interactions near the c and s branch lines in the momentum ranges for which they display cusps, Equation (18). If one goes away from the $(k, \omega)$-plane vicinity of these lines, one confirms that both such a model spectral function and that of the conventional 1D Hubbard model have the broadening discussed in the Supplementary Note 2.

For a short discussion on whether the RPDT is useful to extract information beyond that given by the conventional 1D Hubbard model and corresponding PDT about the physics of quasi-1D metals and a comparison of the PDT and RPDT theoretical descriptions of the line defects in MoSe$_2$, see Supplementary Note 4.

**Data availability.** The data sets generated during and/or analysed during the current study are available from the corresponding authors on reasonable request.

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

## Acknowledgements

The USF group acknowledges support from the National Science Foundation (DMR-1204924). V.K., R.D. and M.-H. P. acknowledges support from the Army Research Office (W911NF-15-1-0626) and thank Prof. Hari Srikanth for resistance measurements in his laboratory. M.C.A., J.A. and C.C. thank enlightening exchanges with Gabriel Kotliar and Zhi-Xun Shen. The Synchrotron SOLEIL is supported by the Centre National de la Recherche Scientifique (CNRS) and the Commissariat à l'Energie Atomique et aux Energies Alternatives (CEA), France. T.Č. and J.M.P.C. thank Eduardo Castro, Hai-Qing Lin and Pedro D. Sacramento for illuminating discussions. The theory group acknowledges the support from NSAF U1530401 and computational resources from CSRC (Beijing), the Portuguese FCT through the Grant UID/FIS/04650/2013 and the NSFC Grant 11650110443.

## Author contributions

Y.M. and H.C.D. contributed equally to this work. They both grew samples by MBE and characterized them by STM. The ARPES data have been obtained and analysed by J.A., H.C.D., C.C. and M.C.A.. The four-point transport measurements have been conducted and discussed by R.D., V.K. and M.H.P. The project has been conceived by M.B. and M.C.A. who directed its experimental part. The theoretical description has been conceived by J.M.P.C. and the corresponding theoretical analysis was carried out by T.Č. and J.M.P.C.. The manuscript has been written by M.B., M.C.A. and J.M.P.C.. All authors contributed to the scientific discussion, contributed to and agreed on the manuscript.

## Additional information

**Competing financial interests**: The authors declare no competing financial interests.

**How to cite this article**: Ma, Y. *et al.* Angle resolved photoemission spectroscopy reveals spin charge separation in metallic MoSe₂ grain boundary. *Nat. Commun.* **8,** 14231 doi: 10.1038/ncomms14231 (2017).

**Publisher's note**: 

