## [Peer Review File · Nature Communications]

Reviewers' comments:

Reviewer #1 (Remarks to the Author):

The manuscript presents evidence for spin-charge separation in one-dimensional metallic system through a combination of angle-resolved photoemission spectroscopy (ARPES), scanning tunneling microscopy (STM), and theoretical modelling. In particular, and what's novel over the previous similar studies, the 1D system considered here is found in mirror twin boundaries of monolayer MoSe₂ films. These line defects can be present in large density in MoSe₂ films, which is relatively easy material to grow, and may thus provide a promising platform for future studies of 1D electron dynamics.

There certainly seems to be two branches in the experimental ARPES data, degenerate near the Fermi-level, but one showing parabolic dispersion and one linear dispersion at lower energies around the Gamma-point. Moreover, the proposed theoretical model reproduces these features, even qualitatively after a renormalization procedure. The model is based on a modification of the pseudofermion dynamical theory (PDT). I am not familiar with the theories on interacting electrons in 1D, and thus

I can not judge whether the model and the renormalization procedure are appropriate.

Nevertheless, there are few remarks and questions that can still be made:

1. There does not appear to be any spectral weight corresponding to the c' branch in the ARPES data. Is there some reason to expect low intensity or is this just a failure of the model? Is it possible to evaluate the ARPES intensity or the spectral broadening from the theory and to compare it to the experiments? The change of exponent from negative to positive is at energies below 1 eV, but ARPES data is cut at 1 eV (corresponding to the position of MoSe₂ VBM) making comparison even more inconclusive. Moreover, ARPES data doesn't show the larger k -values, where c' branch would return to the shown energy window.
2. The dispersions from conventional 1D Hubbard model and PDT model are very similar. In the former case, the exponent is negative for nearly all k -values. However, if the intensity can be very low even for negative exponent, is there really need to go to the PDT theory to explain the findings? It is stated that the α value extracted from the density of states (DOS) is inconsistent with the conventional Hubbard model, but that is assuming that the DOS suppression originates from the electron dynamics according to the TLL theory.

This brings me to another important point. The authors only focus on the agreement between the presented theoretical model and the experiments, but do not consider and rule out alternative explanations for the two bands. Before going further, it is worth considering what would be expected from the quasiparticle picture of these line defects. DFT results should give good indication to this, and, as found in Ref. 4 or Ref. S3, yield one (spin-degenerate) parabolic band turning to linear closer to Fermi-energy and another band closer to the conduction band. The latter is likely above Fermi-level in the present case. That is, the observed features indeed can not be described by DFT (or GW) calculations, and I think this is something that the authors should mention in the manuscript. Also, I suspect that the interaction with the MoS₂ substrate should be small. On the other hand, Fig. 1b shows roughly equal portions of darker and brighter areas. Do these correspond to monolayer and bilayer regions, respectively? If there are two layers with aligned line defects, they could also interact strongly and lead to splitting of the bands. Moreover, from the STM images, the line defects are only about 20 MoSe₂ units long. In addition to the breaking of the translational symmetry, this also leads to quantum confinement (or edge effects) with possible gap opening at the Fermi-level that could explain the DOS suppression.

Even in light of all this, I am actually quite convinced that the observed ARPES features do indeed show the spin-charge separation. The work contains sufficient novelty and can be of wide interest, and thus could deserve publication in Nature Communications once the issues listed above are addressed.

Reviewer #2 (Remarks to the Author):

Although I find the paper by Yujing Ma et al. interesting, I do not recommend it for publication in Nature Communications for the following reasons:

(1) I do not agree with the statement that for the spin-charge separation phenomenon “only qualitative agreements could be obtained between the available systems and theories” (as stated by the Authors of the paper). In fact, Refs. 6 and 16 in the paper give a very detailed description of the spin-charge separation phenomenon. I find these papers far more convincing than the current paper, due to the much cleaner experimental signatures of the spin-charge separation phenomenon: unlike in the current study, the spinon and holon bands are observed in that studies as *clearly* separate features in the ARPES spectra. Moreover, the detailed numerical results unambiguously confirm that the experimentally observed features are indeed separate spinon and holon bands.

In fact, a very closely related situation is observed in a 1D spin-orbital system [1]: also in this case the agreement between theory describing separation of the electron’s quantum numbers in 1D and the (RIXS) experiment is very good on the *quantitative* level (and basically no fitting parameters are used in that study).

(2) Moreover, I find that the current study has the following two (rather important, in my opinion) drawbacks:

(a) In my opinion, nowadays, any study of a correlated system which is to be regarded as being “unambiguous” and of top quality should also be supplemented by numerically exact studies of the models in question [on top of the experimental (ARPES) and the semi-analytical results (PDT) as it is the case here].

(b) One of the crucial cartoons, which should enable the Reader to better understand the problem, Fig. 4(a), is very misleading: as far as I understand it, we have here a doped Hubbard chain (1/3 doping) with holes already introduced to the ground state of the system and (on top of that) it is not clear to me whether the line defect of the studied material has antiferromagnetic correlations. In fact, that the system is doped is one of the main drawbacks of the current study — since it makes this paper far more complicated than e.g. Refs. 6 and 16 and not easily accessible for a general reader of Nature Communications.

(c) Finally, I could not verify whether indeed the line defect in MoSe₂ can be modelled by a Hubbard-like model — this should be better explained in the paper (does the quantum chemistry / DFT calculations predict that the single band extended Hubbard model is enough to describe the low energy physics of these line defects?).

[1] J. Schlappa et al., Nature 485, 82 (2012).

Reviewer #3 (Remarks to the Author):

In my opinion unambiguous detection of spin-charge separation has not been yet achieved in the photoemission studies of one-dimensional conductors. In the present paper the authors are describing their attempt to observe signatures of the Luttinger liquid and spin-charge separation in the photoemission spectra of line defects existing in the single-layer film of MoSe₂ grown on top of MoS₂ single crystal. Idea to examine these objects as possible hosts of Luttinger liquids is very elegant.

So far networks of the line defects in MoS₂ and MoSe₂ have been studied by STM and tunneling

spectroscopy (e.g. PRL 113, 066105; ACS Nano 9, 6619; Nature Materials 12, 554). To the best of my knowledge the present manuscript communicates the first ARPES data collected using thin sample of transition metal dichalcogenide (MoSe₂) with the high density of line defects. The raw data reveal states following parabolic dispersion. Their spectral weight around the Fermi energy is suppressed. Suppression of the spectral weight is indeed expected in the Luttinger liquids. It however also can be a consequence of many factors which are not related to the spin-charge separation. It is therefore not surprising that the authors wish to deliver an "ultimate" evidence of the spin-charge separation. That is the detection of holon and spinon branches which in theory might manifest themselves as distinct dispersing features in the ARPES data. Alas, the data analysis which has yielded the dispersion relies on the fit procedure of a dubious quality. Fit results are summarized in Figure S4. It displays broad photoemission peaks with the long high energy tails. Yet the authors fit them with two Lorentz lines. Quality of such fits is marginal. Assigning two peaks used in the fit procedure to respectively spinon and holon is a pure speculation. Besides, photoemission peaks from the Fermi liquids might have high energy tails (J. Electron Spectrosc. Relat. Phenom 68, 111).

As to the suppression of the spectral weight at the Fermi energy, it alone cannot be taken for the proof of the Luttinger liquid behavior. It for example might be caused by the final state effects similar to those described in Science 284, 777 or PRL 81, 4608.

To conclude, it appears that the experiment did not deliver a solid proof of charge-spin separation. Hence, it must be hard to justify publication of this paper in its present form. However, detection of states originating from the line defects is a significant achievement on its own. Given the current interest to the properties of thin films of transition metal dichalcogenides and their potential applications a better understanding of these states is the must. A comprehensive photoemission and STM study focusing on the geometrical and electronic structures of line defects might well be reported in Nature Communications.

1- REPLY TO THE COMMENTS OF THE REVIEWERS

In the following we reproduce the reviewer's comments in *italic*, our response in **blue-bold**. Since changes to the manuscript were significant and some changes are in response to two or all three reviewers we give a summary of changes to the manuscript in a separate section after the general replies to the reviewers.

OUR REPLY TO THE REMARKS OF REVIEWER #1

Reviewer #1 writes:

The manuscript presents evidence for spin-charge separation in one-dimensional metallic system through a combination of angle-resolved photoemission spectroscopy (ARPES), scanning tunneling microscopy (STM), and theoretical modelling. In particular, and what's novel over the previous similar studies, the 1D system considered here is found in mirror twin boundaries of monolayer MoSe2 films. These line defects can be present in large density in MoSe2 films, which is relatively easy material to grow, and may thus provide a promising platform for future studies of 1D electron dynamics.

There certainly seems to be two branches in the experimental ARPES data, degenerate near the Fermi-level, but one showing parabolic dispersion and one linear dispersion at lower energies around the Gamma-point. Moreover, the proposed theoretical model reproduces these features, even qualitatively after a renormalization procedure. The model is based on a modification of the pseudofermion dynamical theory (PDT). I am not familiar with the theories on interacting electrons in 1D, and thus I can not judge whether the model and the renormalization procedure are appropriate. Nevertheless, there are few remarks and questions that can still be made:

Our Reply:

The initial comments of referee #1 make us realize that we were not enough sharp-witted to present and convince potential readers about the value and novelty of the manuscript. Certainly, as the referee has indicated a notable aspect of our study is the excellent choice of selecting the line defects of MoSe2 as a prototype for 1DES. However, the present work also has accomplished a remarkable novelty in the knowledge of spin-charge separation **in metallic 1D systems**. Compiling, efficiently, high energy- and momentum-resolution ARPES with a powerful theoretical approach, in an extended energy versus momentum space we have been able to reveal the intrinsic microscopic mechanism that governs the quantum electronics of this prototype 1DES. Previous ARPES studies have focused on Mott-Hubbard insulators instead on 1D metals, which relates to a simpler physical problem, see manuscript (ms) references.

We have overcome these initial deficiencies of our manuscript, stating clearly in the present version of the ms and SI, all the achievements procured by our study.

Reviewer #1 writes:

1.-There does not appear to be any spectral weight corresponding to the c' branch in the ARPES data. Is there some reason to expect low intensity or is this just a failure of the model? Is it possible to evaluate the ARPES intensity or the spectral broadening from the theory and to compare it to the

experiments? The change of exponent from negative to positive is at energies below 1 eV, but ARPES data is cut at 1 eV (corresponding to the position of MoSe2 VBM) making comparison even more inconclusive. Moreover, ARPES data doesn't show the larger k-values, where c' branch would return to the shown energy window.

Our Reply:

The referee has identified one of the more relevant by-products of our work: the particular behaviour of the c' branch spectral weight of the ARPES data. Effectively, experimental "s" and "c" branches are nicely reproduced by the calculations, however, as pointed out by the referee no experimental signal has been observed in the momentum-energy space where classical 1D Hubbard model calculations predict the dispersion of the c' branch.

Initially, we thought that was just a lack of our ARPES measurements or a weakness of our theoretical approach. However, we have measured ARPES signal all the way down to 10 eV below the Fermi level and no further photoemission signal was found that could be assigned to the c' branch. Moreover, we have completed the initial photoemission data set by measuring the same ARPES data but changing the polarization of the synchrotron radiation light. Essentially, we have carried out a systematic ARPES study by using right and left circular light as well as vertical and horizontal linear polarization of the excitation incident light in order to rule out efficaciously any matrix element effect on the spectral weight of the detected bands. The ARPES data as a function of the light polarization has confirmed the no-appearance of the c' at any polarization of the incident light and at all different detection geometries. This is a strong experimental indication that the invisibility of the c' branch is an intrinsic property of the studied system.

In order to explain this experimental finding, we have complemented our theoretical study with a more detailed quantitative calculation of the physical quantities (in particular α dependence) that determine the photoemission signal in the full energy versus momentum space. For any α value in the available interval $[\alpha_0, \alpha_{\max}]$, the spectral function has in the vicinity of the branch lines a power-law behaviour controlled by momentum dependent exponents. For the momentum ranges for which such exponents are negative, there are cusps related to a particular branch; i.e. s, c or c', however if the exponents are positives the cusps vanish. In essence, the electron finite-range interactions control the microscopic mechanisms that determine the negative or positive values of the exponents. The increase of the electron finite-range interactions (increasing α) modifies the spectral-function matrix elements of the electron annihilation operators between the ground state and the excited states, for any k value. This mechanism suppresses the singularities associated with the cusps in such branch lines at well-defined k ranges. As justified in the ms and the SI, the agreement between the measured spectral function and the calculations is reached for α values $\in [0.75, 0.78]$. For this α range, the renormalised pseudofermion dynamical theory RPDT approach predicts positive exponents for the power-law associated to the c' branch, and negative exponents for those associated to the s and c branches. Thus these α values reproduce the spectral lines and predict the vanishing of the c' branch. Importantly, this α value is consistent with the α value determined independently from the power law behaviour of the DOS at the Fermi-level according to Tomonaga Luttinger Liquid (TLL) theory.

In summary, adding more ARPES data and theoretical calculations in a precise parameters grid, we have verified that branch c' should not be visible. Effectively, electron finite-range interactions acting in the MoSe2 quantum line defects decrease the overlap within the matrix elements associated with excited states. This, in fact, produces the suppression of the singularities associated with the cusps in the c' branch line at well-defined k ranges. This particular behaviour has been

strikingly predicted and interpreted by our RPDT approach and it is in contraposition to the predictions of the conventional 1D Hubbard model.

Reviewer #1 writes:

2. The dispersions from conventional 1D Hubbard model and PDT model are very similar. In the former case, the exponent is negative for nearly all k -values. However, if the intensity can be very low even for negative exponent, is there really need to go to the PDT theory to explain the findings? It is stated that the alpha value extracted from the density of states (DOS) is inconsistent with the conventional Hubbard model, but that is assuming that the DOS suppression originates from the electron dynamics according to the TLL theory.

Our Reply:

This comments is actually a continuation of the first question of Reviewer #1 and it has been partially answered in the previous section. In fact, Fig. 5 of the current ms shows clearly that the conventional 1D Hubbard model predicts negative exponent in almost the whole energy versus momentum space for the branches s , c and c' . Therefore, in principle, all the bands should be detectable by ARPES. Only RPDT approach including suitable finite-range interactions is able to predict positive exponents for the c' branch and in this way it is the only theoretical approach able to find the microscopic mechanisms behind this behaviour. Also, RPDT disclose the reasons why only branches s and c can be detected experimentally and the c' vanishes.

Moreover, the Reviewer #1 touches here on an important point that is also repeated by Reviewer #3, i.e. our assumption that the observed suppression of the DOS and power law dependence can be described by TLL theory. In the revision we state that there is the possibility of other effects. However, given the 1D metallic nature of the defect lines one certainly would expect TLL theory to apply. That in addition DFT calculations on the MoSe2 line defects cannot describe the observed data reveals that a Fermi-liquid picture does not work. This again supports our picture that the observed suppression of the DOS has a 1D origin. Another important point is that, as mentioned above, only using the exponent $\alpha \in [0.75, 0.80]$ derived from the TLL fitting does reproduce the spectral lines calculated by the RPDT for $\alpha \in [0.75, 0.78]$. Or in other words, two independent measurements do give the same power law behaviour (i.e. we could use the fitting of the spectral lines to the RPDT to find an exponent and compare to the TLL α fitting).

Reviewer #1 writes:

This brings me to another important point. The authors only focus on the agreement between the presented theoretical model and the experiments, but do not consider and rule out alternative explanations for the two bands. Before going further, it is worth considering what would be expected from the quasiparticle picture of these line defects. DFT results should give good indication to this, and, as found in Ref. 4 or Ref. S3, yield one (spin-degenerate) parabolic band turning to linear closer to Fermi-energy and another band closer to the conduction band. The latter is likely above Fermi-level in the present case. That is, the observed features indeed can not be described by DFT (or GW) calculations, and I think this is something that the authors should mention in the manuscript.

Our Reply:

The reviewer is right that the DFT calculations cannot describe the electron removal spectrum. It is however important that the electronic structure in DFT is described by a single parabolic band (see also reply to Reviewer 2) for the RPDT approach to be valid. We agree that this is an important point and we added related relevant information to the manuscript.

Reviewer #1 writes:

Also, I suspect that the interaction with the MoS2 substrate should be small. On the other hand, Fig. 1b shows roughly equal portions of darker and brighter areas. Do these correspond to monolayer and bilayer regions, respectively? If there are two layers with aligned line defects, they could also interact strongly and lead to splitting of the bands. Moreover, from the STM images, the line defects are only about 20 MoSe2 units long. In addition to the breaking of the translational symmetry, this also leads to quantum confinement (or edge effects) with possible gap opening at the Fermi-level that could explain the DOS suppression.

Our Reply:

We have been concerned with opening of a band gap, more for the reason of the discussed Peierls (CDW) transition in 1D metals, for which a band gap ~ 50 meV at 4 K was measured by STS. Because of this, we conducted transport measurements to determine the Peierls transition temperature and made sure our ARPES measurements were taken above this temperature, i.e. at 300 K. As the Reviewer points out, quantum confinement in small segments could potentially also cause a gap. However, although the wire-segments are only 6-10 nm in length they are ~ 10 times longer than the Fermi-wavelength of the metal. Thus we would not expect any measurable band gap opening due to quantum confinement at room temperature. As discussed above, the agreement between TLL $\alpha \in [0.75, 0.80]$ fitting of the DOS and RPDT for $\alpha \in [0.75, 0.78]$ gives some more confidence to describing the suppression of the DOS to TLL.

With respect to the mono- and bi-layer region in the provided STM image, the referee touches another important point. The shown STM image in fact is a sample we have grown for resistance measurements for determining the CDW (metal-insulator) transition temperatures where we assumed it would be desirable to have a continuous film, which made the formation of bi-layers unavoidable. For ARPES measurements, we were concerned like the Reviewer that the bilayer regions may modify the electronic structure and therefore we used sub-monolayer MoSe2 samples. STM images of samples used in ARPES are now shown in the SI. These samples are predominantly monolayer.

Reviewer #1 writes:

Even in light of all this, I am actually quite convinced that the observed ARPES features do indeed show the spin-charge separation. The work contains sufficient novelty and can be of wide interest, and thus could deserve publication in Nature Communications once the issues listed above are addressed.

Our Reply:

We are happy that the Reviewer agrees that the observed ARPES features show spin-charge separation. We believe we have addressed here and in the new materials added to the revised manuscript all the issues mentioned by this Reviewer.

OUR REPLY TO THE REMARKS OF REVIEWER #2

Reviewer #2 writes:

Although I find the paper by Yujing Ma et al. interesting, I do not recommend it for publication in Nature Communications for the following reasons:

*(1) I do not agree with the statement that for the spin-charge separation phenomenon “only qualitative agreements could be obtained between the available systems and theories” (as stated by the Authors of the paper). In fact, Refs. 6 and 16 in the paper give a very detailed description of the spin-charge separation phenomenon. I find these papers far more convincing than the current paper, due to the much cleaner experimental signatures of the spin-charge separation phenomenon: unlike in the current study, the spinon and holon bands are observed in that studies as *clearly* separate features in the ARPES spectra. Moreover, the detailed numerical results unambiguously confirm that the experimentally observed features are indeed separate spinon and holon bands.*

*In fact, a very closely related situation is observed in a 1D spin-orbital system [1]: also in this case the agreement between theory describing separation of the electron’s quantum numbers in 1D and the (RIXS) experiment is very good on the *quantitative* level (and basically no fitting parameters are used in that study).*

Our Reply:

In the original manuscript we had already distinguished the references that quoted the **quasi-1D metals** from those that quoted the **quasi-1D Mott-Hubbard insulators (MHI)**, stating that for the latter qualitative agreement had been obtained between these systems and theories. Our study refers to a 1D metal. Following the Reviewer criticism, we agree that the results on quasi-1D Mott-Hubbard insulators give a very detailed description of the spin-charge separation phenomenon, however, these findings can not be applied directly to quasi-1D metals, where the physical scenario is more complicated, as we now describe in the current version of the ms and SI .

Our work is tackling 1D metals that are governed by different physics and thus cannot be compared to the *spin-charge separation phenomenon* observed in MHI. It was our mistake not to draw this distinction clearly in our initial submission. In the revision we emphasize the success of previous ARPES investigations of quasi 1D Mott-insulators and clearly state the difference to our system. (See the introduction of the revised manuscript.) Another point we want to make here is with respect to our quantitative fitting of the data. We are not quite sure if the Reviewer wants to imply that our fitting requires free parameters. If he/she does, we would like to make clear that all the

parameters used in the RPDT have been determined in the experiment and there are no free fitting parameters.

Reviewer #2 writes:

(2) Moreover, I find that the current study has the following two (rather important, in my opinion) drawbacks:

(a) In my opinion, nowadays, any study of a correlated system which is to be regarded as being “unambiguous” and of top quality should also be supplemented by numerically exact studies of the models in question [on top of the experimental (ARPES) and the semi-analytical results (PDT) as it is the case here].

Our Reply:

First, as mentioned in the revised manuscript and described in the SI, the exponent curves provided in the Figs. 5 (a), (b) and (c) and new Figs. S5 (a), (b) and (c) have been obtained from numerical computations needed to solve involved coupled integral equations. Such equations have no analytical solutions. Their numerical solution is though exact. Second, density functional theory (DFT) numerical calculations of the present MoSe₂ quantum line defects have already been performed in references 5 and 34. As discussed in the revised manuscript, the observed features cannot be described by such numerical DFT calculations. This reveals that the MoSe₂ quantum line defects physics is not that of the Fermi-liquid quasiparticles. Last but not least, numerical studies of 1D systems by means of the exact time-dependent density matrix renormalization group (t-DMRG) method require an explicit expression for the Hamiltonian of the quantum problem under consideration. As discussed in the SI, our renormalisation scheme refers to some 1D Hamiltonian with the same terms as the 1D Hubbard model plus finite-range interaction terms. The latter terms are neither a mere first-neighbouring V term nor a complete long-range Coulomb potential extending over all lattice sites. They actually account for the effects of the electron finite-range interactions associated with α values in the small interval $\alpha \in [0.75, 0.78]$. Importantly, the specific form of the additional finite-range interaction Hamiltonian terms is not needed for our study. This follows from the type of universality used in our study implying that both for the low-energy TLL and well-defined energy windows near the high-energy branch lines singularities of the 1D Hubbard model with finite-range interactions associated with any α value in the available interval $[\alpha_0, \alpha_{\max}]$ defined in the revised manuscript the relation of the parameter α to the phase shifts remains exactly the same as for the $\alpha = \alpha_0$ 1D Hubbard model. However, a numerically exact study of the problem by t-DMRG would require the explicit unknown form of the long-range interaction Hamiltonian terms. The advantage of our method is that near the relevant spectral features that information is not needed. This is why the theoretical many-electron problem posed by the present experimental data is very complex and cannot be solved by the presently known methods other than our scheme, which relies on the universality of a well-defined set of quantum problems corresponding to different α values in interval $[\alpha_0, \alpha_{\max}]$. The results then follow from the exact Bethe-ansatz solution of the $\alpha = \alpha_0$ 1D Hubbard model, which is the reference model of the set of universal models under consideration that has an exact solution.

Reviewer #2 says:

(b) One of the crucial cartoons, which should enable the Reader to better understand the problem, Fig. 4(a), is very misleading: as far as I understand it, we have here a doped Hubbard chain (1/3 doping) with holes already introduced to the ground state of the system and (on top of that) it is not clear to me whether the line defect of the studied material has antiferromagnetic correlations. In fact, that the system is doped is one of the main drawbacks of the current study — since it makes this paper far more complicated than e.g. Refs. 6 and 16 and not easily accessible for a general reader of Nature Communications.

Our Reply:

We agree that the cartoon numbered 4 (a) in the original manuscript does not apply to the present case of a 1D metal and thus is misleading. Consequently, we removed it from the revised manuscript. However, we disagree with the Reviewer's suggestion that the fact that the system is doped is a main drawback of our study. In fact we think it is the opposite: Because we solved the problem for a **1D metal with electron finite-range interactions** rather than the already known problem of a **quasi-1D Mott-Hubbard insulator**, enhances the scientific importance of our results. We would agree that in higher-dimensional problems different from ours such as high-Tc superconductivity, doping a Mott-Hubbard insulator, as for instance within the Hubbard model on a square lattice at half-filling, is an extremely complex quantum problem that has remained unsolved despite the many attempts to study it in the last 30 years. In contrast, the 1D Hubbard model has an exact solution both for its Mott-Hubbard insulating phase at electronic density $n=1$ and for the metallic phase for densities $n<1$ and $n>1$. It is the exact solution of the latter 1D quantum problem that, relying on the type of universality used in our study, has allowed the quantitative description of the line defects in MoSe₂ experimental data by the 1D Hubbard model with finite-range interactions corresponding to α values in the small interval $\alpha \in [0.75, 0.78]$. Surprisingly, after the renormalisation that follows from the universality of a well-defined set of quantum problems, the physics is qualitatively similar to that of the well-known 1D Hubbard model exact solution.

With respect to the Reviewer's comment about the accessibility for the readership of Nature Communication, we disagree that the manuscript does not provide important and stimulating read to a general condensed matter physicist or even materials scientist. On the one hand, there is the experimental part that demonstrates ARPES measurements on line defects for the first time. It includes a brief discussion on general properties of these grain boundaries of MoSe₂ like TLL-DOS and Peierls transition, which is easily accessible. In addition, there is the main point about spin-charge separation, which has been a corner stone in 1D materials and the demonstration of it in 1D metallic line defects is of general interest. On the other hand, the details of the RPDT theory may be beyond the interest of most 'general' readers, but then again Nature Communications, in contrast to other Nature X journals, does also publish work of significant scientific advancements that are of interest for specialists in certain fields. Our manuscript attempts, we think successfully, to have the main text as accessible to a broader audience. Most of the more theoretical details are provided in the Supplementary Information for the 'specialists'. We also should mention that when talking about this subject at conferences, it always receives strong interest by people who are not specialists in PDT. One would expect similar interest in this paper.

Reviewer #2 writes:

(c) Finally, I could not verify whether indeed the line defect in MoSe₂ can be modelled by a Hubbard—like model — this should be better explained in the paper (does the quantum chemistry / DFT

calculations predict that the single band extended Hubbard model is enough to describe the low energy physics of these line defects?).

Our Reply:

The existence of detailed results provided in references 5 and 34 of the revised manuscript of previous DFT simulations of line defects in MoSe₂ justifies why such studies have not been repeated here. The figure below has been taken from ACS Nano 9, 3274-3283 (2015), and shows that the defect structure is characterized by a single band (note that the green band is due to the edges of the simulated ribbons and thus do not apply to us, only the orange band is relevant to our system and only the lower branch is below the Fermi-level). As already pointed out by Reviewer #1, the upper band in the DFT model is above the Fermi-level. Thus DFT calculations indicate that the single band extended Hubbard model is suitable to describe these line defects.

Figure 3. Atomic and electronic structures and density of states of the MTB. (a) The band structure and (b) corresponding density-of-states from a ribbon calculation with one MTB. The states are colored by the projection to different regions of the system (orange, near MTB region; green, edge region; black, bulk region). (c–e) The atomic structure of the MTB system together with the charge density isosurface visualization of wave functions of the two MTB-localized states (i) and (ii) within the gap.

Reviewer #2 writes:

[1] J. Schlappa et al., *Nature* 485, 82 (2012).

Our Reply:

We have added this reference to the revised manuscript and have shortly commented in it the interesting type of separation observed in the quasi-1D Mott-Hubbard insulator under consideration in that paper.

OUR REPLY TO THE REMARKS OF REVIEWER #3

Reviewer #3 says:

In my opinion unambiguous detection of spin-charge separation has not been yet achieved in the photoemission studies of one-dimensional conductors.

In the present paper the authors are describing their attempt to observe signatures of the Luttinger liquid and spin-charge separation in the photoemission spectra of line defects existing in the single-layer film of MoSe₂ grown on top of MoS₂ single crystal. Idea to examine these objects as possible hosts of Luttinger liquids is very elegant.

So far networks of the line defects in MoS₂ and MoSe₂ have been studied by STM and tunneling spectroscopy (e.g. PRL 113, 066105; ACS Nano 9, 6619; Nature Materials 12, 554). To the best of my knowledge the present manuscript communicates the first ARPES data collected using thin sample of transition metal dichalcogenide (MoSe₂) with the high density of line defects. The raw data reveal states following parabolic dispersion. Their spectral weight around the Fermi energy is suppressed. Suppression of the spectral weight is indeed expected in the Luttinger liquids. It however also can be a consequence of many factors which are not related to the spin-charge separation. It is therefore not surprising that the authors wish to deliver an “ultimate” evidence of the spin-charge separation. That is the detection of holon and spinon branches which in theory might manifest themselves as distinct dispersing features in the APRES data.

Alas, the data analysis which has yielded the dispersion relies on the fit procedure of a dubious quality. Fit results are summarized in Figure S4. It displays broad photoemission peaks with the long high energy tails. Yet the authors fit them with two Lorentz lines. Quality of such fits is marginal. Assigning two peaks used in the fit procedure to respectively spinon and holon is a pure speculation. Besides, photoemission peaks from the Fermi liquids might have high energy tails (J. Electron Spectrosc. Relat. Phenom 68, 111).

As to the suppression of the spectral weight at the Fermi energy, it alone cannot be taken for the proof of the Luttinger liquid behavior. It for example might be caused by the final state effects similar to those described in Science 284, 777 or PRL 81, 4608.

To conclude, it appears that the experiment did not deliver a solid proof of charge-spin separation. Hence, it must be hard to justify publication of this paper in its present form.

However, detection of states originating from the line defects is a significant achievement on its own. Given the current interest to the properties of thin films of transition metal dichalcogenides and their potential applications a better understanding of these states is the must. A comprehensive photoemission and STM study focusing on the geometrical and electronic structures of line defects might well be reported in Nature Communications.

Our Reply:

First, we thank the Reviewer for pointing out that there indeed has not been any previous unambiguous detection of spin-charge separation in 1D metals and for finding our measurements ‘elegant’. However, we strongly disagree with the assertion that there is no strong experimental evidence for a split band with two distinct dispersions. While the fitting of the raw data is somewhat delicate, though clearly not fittable with a single component, the standard approach

using second derivatives as described in ref. 35 of the revised ms gives clear evidence of the two dispersing bands. It is the dispersion resulting from the latter data analysis procedure (without any peak fitting that may be considered as 'dubious') that is in exact agreement with the prediction from the RPDT. At this point we have to emphasize that the band dispersions from that RPDT does not have any free fitting parameters, but rather the band filling, transfer integral and the effective electron interaction are determined from the ARPES data and thus the exact agreement is unlikely to be coincidental.

The reference given by the Reviewer that is supposed to give an alternative explanation for the asymmetric peak does not apply to our study. The reference is a study on a Fermi-liquid, but more importantly the photoemission peaks studied in that work are photoemission intensity directly at the Fermi-level and thus the asymmetry in the peaks is partially a consequence of the cut-off at the Fermi-level. In our case the asymmetric 'broadening' is observed at the bottom of the band $\sim 1\text{eV}$ away from the Fermi level.

Most importantly, any experimental technique would 'label' a spectral line as spinon or holon line does not exist and thus there is no unambiguous measurement one could conduct. In fact we believe the best 'proof' is the combination that the observed features cannot be described by numerical DFT calculations, which reveals that the MoSe₂ quantum line defects physics is not that of the Fermi-liquid quasiparticles, and the agreement with 1D RPDT. Furthermore, the Reviewer is right in pointing out the alternative explanations for the suppression of the DOS at the Fermi level which we now introduce a brief paragraph that acknowledged such alternative explanations but also point out work on other (quasi) 1D metals that use TLL to explain this behaviour.

We are also very pleased with the Reviewer for acknowledging the importance of careful characterization of the defect structures in MoSe₂ and that we are the first to do so by ARPES. This (and the lessened restriction to the manuscript-length), encouraged us to move some of the material concerning the CDW transition from the supplement to the main manuscript, to provide a complete description of these 1D grain boundaries. As pointed out, the CDW transition in these defects has been previously reported in Nature Physics by low-T STM/STS studies. The observation of charge density periodicity of 3 times the lattice constant is in agreement with our low-T STM work. However, in the previous work there was no justification for this periodicity. Our ability of doing ARPES on these defect lines provides an elegant answer by directly measuring the nesting vector. Also our transport measurements enabled us to determine the CDW transition temperature(s) -- another critical property of these line defects. Adding these measurements to the main manuscript, we present a more complete characterization of the twin grain boundaries in MoSe₂, what we think is in the spirit of the Reviewer's comments. To emphasis this change to the manuscript, we also changed the title to '***Angle resolved photoemission spectroscopy measurements on quantum line defects in MoSe₂: Evidence for spin charge separation***' and modified the abstract significantly.

Finally, we have quoted in the revised manuscript most of the references mentioned in the remarks of Reviewer #3.

REPLY TO THE COMMENTS OF THE REVIEWERS

As in our first reply letter, in the following we reproduce the Reviewer's comments in *italic* and our response in **blue**. Since changes made to the manuscript were not significant, we briefly list at the end of the reply the changes included in the present version of the manuscript.

OUR REPLY TO THE REMARKS OF REVIEWER #1

Reviewer #1 writes:

The authors have given satisfactory responses to the comments and criticism presented in my previous report, and modified the manuscript accordingly. I still think that publication in Nature Communications is justified. My only concern is that it seems that none of the Reviewers went, or were able to go, through all the details of the theory presented in the SI.

Minor remark:

The caption of Fig.S5 purports to show results for $\alpha=0.75$, but there is no such line shown or the legend is incorrect.

Our Reply:

We certainly appreciate the judgement of the Reviewer #1 that has comprehended and acknowledged our work and explanations.

Moreover, we have corrected the misprints in the caption of Figure S5.

Reviewer #2 writes:

I would like to thank the Authors for addressing my criticism in such a great detail and explaining to me all the concerns that I had after reading the first version of the paper. Therefore, I can now recommend the paper for publication in Nature Communications.

Our Reply:

We also appreciate the very positive comments of Reviewer #2 as well as his favourable advice concerning the publication in Nature Communication of our work in its present form. Moreover, we would like to thank for his incisive and constructive previous criticisms. Clarification of his/her questions included the addition of new details of our theoretical analysis to the SI. That he/she considered that we addressed his/her questions in great detail and thanked us for explaining the corresponding details, ensures that he/she went through all previous and new details of the theory and the importantly the stunning agreement between the theoretical prediction with the experimental results.

Reviewer #3 writes:

1.- When it comes to reporting photoemission data indicating an "unambiguous" Luttinger liquid behavior along with the spin-charge separation one might expect that these data will be thoroughly scrutinized and every aspect of the data will be checked against predictions. So far the authors have only looked at the dispersion relations. They do believe that dispersions extracted from the photoemission data agree reasonably well with the predictions of the theory and that such agreement signals existence of the Luttinger liquids in the one-dimensional line defects. In my opinion such agreement does not necessarily prove Luttinger liquid behavior.

Our Reply:

We agree with the Reviewer that for proving the occurrence of spin-charge separation behaviour in 1DEG metals, the full data set should be carefully scrutinized. In fact we have done such analysis before and following the suggestions of the reviewer we have now presented it in a new SI section.

Effectively, as described in the new SI section, following a throughout analysis of EDC (Energy Distribution Curve) and MDC (Momentum Distribution Curve) as the Reviewer has requested, we have found spin-charge separation signatures and even fractionalization signatures. This is complementarily to the conclusive and remarkable ARPES agreement with the theoretical predictions based on 1D Hubbard with finite-range interaction approach that is presented in the manuscript and whose details are also provided in the SI.

We take issue with the Reviewer's assertion that we only compare band dispersions derived from the 1D Hubbard model with finite range interactions with the experimental data. In fact the agreement between our theory and experiment goes beyond 'just' the agreement of the dispersion relations. Some parts of the theoretical dispersion spectra are not seen. And we have predicted theoretically which parts are seen and which parts are not seen. Within a Lehmann representation, the spectral function displays the spectra and the spectral weights. The latter are encoded within matrix elements of the electron operators between initial and final energy eigenstates. In addition to the corresponding dispersion spectra, we have studied the theoretical quantum overlaps associated with such spectral function weights. The matrix elements quantum overlaps lead to singular cusps in the (k, ω) plane that coincide with the cusps observed in the experiment. Hence in contrast to what is said by the Reviewer, our study reveals agreement between the 1D Physics theory and the experiments both in what the dispersion relations and spectral weights is concerned.

Reviewer #3 writes:

2.- In the first report I have outlined a concern regarding fitting photoemission lines with two peaks. The concern is still there. Energy distribution curves (aka raw photoemission data) are the broad peaks with the long high-energy tails. They do not show any obvious splitting, which might have suggested presence of two peaks corresponding to spinon and holon branches (to the contrary, lines showing results of the fitting procedure do show a very clear splitting). Nature of the high-energy tails could be debated. It is possible to assume that they arise due to the spin-charge separation. However, in order to put this assumption on the solid ground the authors should go further and examine other aspects of the data. They could for instance look into the lifetimes and check how they depend on energy and temperature (in the Fermi liquids they follow quadratic dependence). The authors might adhere to the usual practice and analyze the momentum distribution curves, which are commonly used for extracting dispersion relations and lifetimes. I cannot recommend publishing this paper in its present form.

Our Reply:

There are a few points raised regarding the raw data analysis of the photemission spectra. These are in principle valid points, however, as we discuss below these proposed analyses are less solid and there are fundamental reasons for broadening of the spectra in 1D metals as well for the observation of a continuum rather than well-defined spinon and holon bands. All these observations are consistently explained in the 1D Hubbard model with finite range interactions.

Specific replies to the reviewer's main points are:

1.- Broadening of the ARPES spectral function: It is well-know that in 1D metallic systems there

are no quasi-particles in the vicinity of the Fermi surface. The excitations rather are gapless collective modes involving charge and spin degrees of freedom. All these features have dramatic consequences, particularly on the ARPES directly measured spectral function of interacting 1D fermions. Specifically, any structure of the spectral function other than the Fermi-energy cut-off, is noticeably very broad. This is another typical hallmark of spin-charge separation due to the fact that there are no stable excitations with the quantum numbers of the electron. Therefore, on the contrary to what the Reviewer suggests concerning the broad nature of the peaks, we are rather pleased that they cannot be fitted only with two peaks, corresponding to the spinon and the holon branches, respectively. This broadening occurs both at low energy near the Fermi level and at higher energies. Effectively, this broadening of the photoemission peaks is a hallmark of the spin-charge separation; as already it has been reported (see Fig. 5 of ref. 17 of our manuscript, for more details) by ARPES groups as prestigiously as the Stanford and IOP Beijing groups, in particular, Prof. Z.-X Shen and Prof. X. J. Zhou, respectively.

Figure 1. (a) Raw ARPES data, (b) second derivative of data in panel (a), (c) schematic description between the EDC shape and the lifetime, (d) MDC plots at different binding energies extracted from panel (a) data and finally panel (e) shows EDC plots at different momentum (k) values indicated in panel (a) as yellow straight lines.

2.- Analysis of the EDC, MDC and lifetime: The lifetime of the quasiparticle, $\tau(k)$, can be directly determined from the width of the peak in the EDC, analysing the ARPES data defined by the spectral weight at fixed k as a function of ω , where ω is the energy, therefore $1/\tau = \Delta\omega$ (eq. 1). The

consistency of a Fermi liquid picture can be also obtained by studying the MDC, i.e., by studying the width Δk of the spectral function peak at fixed binding energy, ω . As long as the quasi-particle excitation is well defined, (i.e., the decay rate is small compared to the binding energy), these two widths are related by: $\Delta\omega = v_F \Delta k$ (eq. 2), where v_F is the renormalized Fermi velocity, which has been directly measured using high energy and momentum resolution ARPES. All these concepts have been robustly probed in the early ARPES time, see for more details; T. Valla et al., Phys. Rev. Lett. 83, 2085 (1999).

Due to the separation of charge and spin, one hole (or one electron) is always unstable to decay into two or more elementary excitations, of which one or more carries its spin and one or more carries its charge. Consequently, the ARPES spectral function does not have a pole contribution, but rather consists of a ***multiparticle continuum***. If both the spin and charge excitations are gapless, elementary kinematics implies that, at $T=0$, the spectral function is nonzero only for negative frequencies such that $|\omega| \geq \min(v_c, v_s)|k|$ (eq. 3), where v_c and v_s are the charge and spin velocity, respectively. This behaviour can be observed in the raw ARPES data of Fig. 1a, where the spectral function particularly at ω values between 0.40 eV up to 0.95 eV shows a continuum, which is valid for all momentum (k) values that fit equation (3). Thus contrary to the reviewer suggestion, there should be no ‘splitting’ of the spectral function, but as we outline in the manuscripts only cusps of the spectral function should be observable in agreement with our data. The occurrence and dispersion of these cusps are in quantitative agreement with our theory.

As the Reviewer #3 has suggested, MDC and EDC plots are sensitive to this detachment of the system with respect to a conventional Fermi-liquid quasiparticle behaviour. In fact, this type of analysis, based on the shape of EDC and MDC plots, has already been published a long time ago by Emery et al. , (see Fig. 2 and Fig. 3 in ref. 5 of our manuscript). In Figure 1 we present the results of a similar analysis. As shown in panels (d) and (e) the raw data MDC and EDC cuts at different binding energies and momentum, respectively, show a clear enlargement of the lifetime that can be extracted from the ARPES data. However, this experimental value is just ***proportional*** to various interaction strengths. Consequently, the evaluation of the nature and strength of the interactions that we can be extracted from this analysis is just approximate compared with the state-of-the-art theoretical approach reported in the present manuscript.

3.- Remarks on temperature dependent studies. Also Reviewer #3 is suggesting temperature dependent studies, which ideally we agree could give possibly some additional, albeit non-critical, information. As pointed out in our manuscript it is only meaningful to do studies of the Luttinger liquid above the Peierls’ (charge density wave) transition (i.e. above ~ 235 K) and thus this precludes a meaningful analysis of temperature dependent data, without a convolution of effects. Another technical complication with low temperature analysis arises from the semiconducting nature of our substrate (MoS_2). While this ensures that our 1D electronic states to lie entirely within the band gap and thus avoid band overlap, which has resulted in misinterpretation of 1D states in the past, it has the downside of causing space charge effects at low temperatures, effectively making a meaningful low temperature photoemission study impossible.

The discussion of line broadening and lifetimes and the consistent explanation of the spectral features are now also included in the revised Supplementary Information under the new Section 6.

Reviewers' Comments:

Reviewer #3 (Remarks to the Author)

The authors did a very good job revising the manuscript and appending new information which shall allow motivated readers to navigate through the published studies of spin-charge separation and judge the quality of the data and discussion put forth by the authors. At this point the manuscript looks publishable to me.